JCB Journal of Cell Biology

# The C10orf76–PI4KB axis orchestrates CERT-mediated ceramide trafficking to the distal Golgi

Aya Mizuike[1,2], Shota Sakai[1], Kaoru Katoh[3], Toshiyuki Yamaji[1], and Kentaro Hanada[1,2]

Phosphatidylinositol 4-monophosphate [PtdIns(4)P] is a precursor for various phosphoinositides but also a membrane-embedded component crucial for membrane contact sites (MCSs). Several lipid transfer proteins are recruited to MCSs by recognizing PtdIns(4)P; however, it remains poorly elucidated how the production of PtdIns(4)P for lipid transport at MCSs is regulated. Following human genome-wide screening, we discovered that the PtdIns(4)P-related genes *PI4KB*, *ACBD3*, and *C10orf76* are involved in endoplasmic reticulum-to-Golgi trafficking of ceramide by the ceramide transport protein CERT. CERT preferentially utilizes PtdIns(4)P generated by PI4KB recruited to the Golgi by C10orf76 rather than by ACBD3. Super-resolution microscopy observation revealed that C10orf76 predominantly localizes at distal Golgi regions, where sphingomyelin (SM) synthesis primarily occurs, while the majority of ACBD3 localizes at more proximal regions. This study provides a proof-of-concept that distinct pools of PtdIns(4)P are generated in different subregions, even within the same organelle, to facilitate interorganelle metabolic channeling for the ceramide-to-SM conversion.

## Introduction

The emergence of membrane-bound organelles appears to have been a key event in the evolution of life, which enabled the manipulation of numerous metabolic reactions as well as macromolecule functions in the cell. Lipids are essential constituents of organelle membranes, and the lipid composition of membranes has a major impact on organelle structure and function (Van Meer et al., 2008; Casares et al., 2019; Mironov et al., 2020). The ER, which comprises ~70% of the total membranes in a cell, serves as a major synthetic site for a variety of types of lipids (Jacquemyn et al., 2017). The subregions of an organelle are often in close proximity (10–30 nm) to other organelles, forming membrane contact sites (MCSs; Jain and Holthuis, 2017; Prinz et al., 2020). The ER is known to communicate with almost all of the other organelles (e.g., mitochondria, Golgi complex, and plasma membrane [PM]) via MCSs (Wu et al., 2018; Almeida and Amaral, 2020). Non-vesicular lipid transfer mediated by lipid transfer proteins (LTPs) mostly occurs at MCSs, enabling the effective and accurate transport of lipids (Hanada, 2018; Balla et al., 2019; Wong et al., 2019; Jain and Holthuis, 2017). However, it remains poorly elucidated how a specific LTP works at a specific MCS. In addition, it is unclear how different subregions are generated within the same organelle to contact with the ER.

At the MCSs formed at the interface of the ER and the Golgi apparatus, several LTPs transfer lipids, indirectly coupling with lipid metabolism (Hanada, 2018; Mesmin et al., 2019; Goto et al., 2020; Venditti et al., 2020). CERT, a ceramide transfer protein, dually targets the ER and Golgi membranes at the MCSs via interactions with the ER-resident proteins VAP-A or VAP-B and with Golgi-embedded phosphatidylinositol 4-monophosphate (PtdIns(4)P; Hanada et al., 2003; Kawano et al., 2006). Ceramide transferred by CERT is converted to sphingomyelin (SM) by the SM synthases SMS1 and SMS2, mainly at the trans-cisternae of the Golgi apparatus or at the TGN. Among the several functional roles of phosphoinositides (Di Paolo and De Camilli, 2006; Hammond and Burke, 2020), one of the main roles of Golgi-embedded PtdIns(4)P is to anchor LTPs working at the Golgi apparatus through interactions with the pleckstrin homology (PH) domain (Tóth et al., 2006; Balla et al., 2005). PtdIns(4)P also serves as a driving force for cholesterol transfer by oxysterol binding protein (OSBP) against the uphill concentration gradient, consuming an equimolar PtdIns(4)P molecules for cholesterol transport (Mesmin et al., 2013). Replenishment of PtdIns(4)P is supported by phosphatidylinositol (PtdIns) supplementation to the Golgi apparatus by PtdIns transfer proteins such as Nir2 (Peretti et al., 2008).

[1]Department of Biochemistry and Cell Biology, National Institute of Infectious Diseases, Tokyo, Japan; [2]Department of Quality Assurance, Radiation Safety and Information System, National Institute of Infectious Diseases, Tokyo, Japan; [3]Biomedical Research Institute, National Institute of Advanced Industrial Science and Technology (AIST), Ibaraki, Japan.

Correspondence to Kentaro Hanada: hanak@niid.go.jp, hanaknih@gmail.com.



The human genome encodes several types of PtdIns 4-kinases for the synthesis of PtdIns(4)P from PtdIns (Boura and Nencka, 2015; Burke, 2018; Dickson and Hille, 2019). In various cell types, PI4KB (also known as PtdIns 4-kinase IIIβ) represents the major kinase that produces PtdIns(4)P in the Golgi apparatus (Hausser et al., 2005; Balla et al., 2005), while PI4K2A (also known as PtdIns 4-kinase IIα), which mainly works in endocytic compartments and the TGN, also contributes to PtdIns(4)P production in the Golgi complex (Balla et al., 2002; Minogue et al., 2010). As PI4KB is a cytosol-soluble protein in nature, it has been suggested that PI4KB is recruited to the Golgi apparatus via interactions with Golgi-associated factors (Sasaki et al., 2012; Daboussi et al., 2017).

In this study, to identify the genes required for CERT to properly function at the ER–Golgi MCSs, we carried out genome-wide CRISPR knockout (KO) screening in HeLa cells using the SM-binding cytolysin lysenin (Yamaji et al., 1998). We identified 93 genes, the disruption of which likely confers lysenin resistance to HeLa cells. Among the proteins encoded by these genes, we focused on PI4KB and its interacting proteins ACBD3 and C10orf76. We anticipated these proteins would play crucial roles in the production of PtdIns(4)P in Golgi subregions, where CERT catalyzes the ER-to-Golgi transfer of ceramide. Analysis of the KO strains suggested that ACBD3 and C10orf76 are able to work independently in the generation of the PtdIns(4)P pools required for the Golgi-anchoring of CERT, via the recruitment of PI4KB to the Golgi apparatus. Observation with super-resolution microscopy, with which the resolution was sufficient to discriminatively resolve the cis-Golgi and the TGN, revealed that the intra-Golgi distribution of ACBD3 and C10orf76 is mostly separate, although they partially overlap. We also succeeded in observing the heterogenous distribution of endogenous VAP-A within the ER, some of which was in close vicinity or overlapped with CERT. We here propose that among the diverse PtdIns(4)P pools generated in the Golgi complex, the C10orf76-dependent pool formed at the limited subregion of Golgi adjacent to the ER recruits CERT to the distal Golgi regions, thereby facilitating the metabolic flow of ceramide to be converted to SM.

## Results

### Human genome-wide KO screening to identify genes conferring lysenin resistance

In a previous study, we showed that the selection of chemically mutagenized Chinese hamster ovary (CHO) cells by the SM-binding cytolysin lysenin resulted in mutant CHO cell lines defective in SM synthesis. However, the repertoire of selected mutants obtained using the conventional somatic cell genetics approach was limited to CERT and the SPTLC1 subunit of serine palmitoyltransferase (Hanada et al., 1998; Hanada et al., 2003). We reasoned that the selection of lysenin-resistant variants from HeLa mutant cell libraries affected by genome-wide CRISPR/Cas9 KO (Sanjana et al., 2014; Yamaji et al., 2019) would allow us to conduct a more comprehensive selection of candidate genes. Lysenin binds SM clustered in the exoplasmic leaflet of the PM (Ishitsuka et al., 2004; Kiyokawa et al., 2005), therefore candidate genes were presumed to encode proteins

involved in SM synthesis, delivery, or intra-PM distribution (illustrated in Fig. 1 A); this should include those proteins involved in the formation and/or maintenance of MCSs critical for CERT. Thus, we exposed the library cells to a low dosage of lysenin three times, expecting to obtain mutant cells of various levels of lysenin resistance (Fig. 1 B). From our analysis of the enrichment of single-guide RNAs (sgRNAs) in the lysenin-resistant cell pools relative to the lysenin-nonexposed control pools, the genes we selected as candidates were genes for which the sgRNAs were reproducibly enriched more than fivefold.

We identified 93 genes by this screening process, which could be classified into several categories: SM synthesis, phosphatidylserine (PtdSer) metabolism, PtdIns(4)P metabolism, Golgi structure, membrane traffic, signal transduction, and transcriptional/translational regulation (Fig. 1 B, Fig. S1, and Table S1). The sgRNAs of genes involved in the synthesis of SM (e.g., SPTLC1, SGMS1, SGMS2, and CERT1) were comprehensively enriched (Fig. 1, A–C), which validated the effectiveness of this genome-wide screening approach. Membrane traffic-related genes identified by the screening may participate in SM delivery to the PM or retention of lipid synthetic enzymes at proper sites; in addition, it is noteworthy that components of the exocyst complex and the COG complex were enriched in the screening (Fig. S1). As for PtdSer-related genes, genes encoding a PtdSer synthase, a PtdSer lipase, and a β-subunit (known as TMEM30A or CDC50A) of P4-ATPases were obtained (Fig. S1). This may support a recent model where it was proposed interleaflet coupling between PtdSer with pairs of specific acyl chains and very-long chain SM leads to co-sorting of these lipids within the membrane (Skotland and Sandvig, 2019). Compared with the genes involved in de novo SM synthesis, most of the other genes identified by the screening exhibited lower enrichment ratios. Nevertheless, PI4KB (also known as PI4KIIIβ, which encodes PtdIns 4-kinase IIIβ) was highly enriched. Moreover, ACBD3 (alias GCP60) and C10orf76 (alias ARMH3), both of which encode PI4KB-binding proteins (Greninger et al., 2013; Klima et al., 2016; McPhail et al., 2020), were also enriched, supporting the potential involvement of PI4KB in lysenin-sensitivity.

### PI4KB, ACBD3, and C10orf76 as PtdIns(4)P metabolism-related genes

To explore how the generation of PtdIns(4)P that recruits CERT to function at the ER-Golgi contacts is regulated, we hereafter focused on the category of PtdIns(4)P metabolism among the above categories. The candidate genes in this category included PI4KB, ACBD3, and C10orf76. While two PtdIns 4-kinases PI4K2A (also known as PI4KIIα) and PI4KB are responsible for the generation of PtdIns(4)P pools within the Golgi complex, only sgRNAs of PI4KB were concentrated in the screening (Fig. 1 C), in line with reports that the knockdown or pharmacological inhibition of PI4KB represses CERT-dependent SM synthesis (Tóth et al., 2006; Capasso et al., 2017).

Among the reported PI4KB-binding proteins, ACBD3 and C10orf76 were the only candidates obtained from the screening. ACBD3, a member of the ACBD protein family (reviewed in Islinger et al., 2020), is a multi-domain protein that contains an acyl-CoA binding (ACB) domain (which mainly binds palmitoyl-

Figure 1. **Identification of genes conferring lysenin resistance by disruption. (A)** Pathways involved in de novo SM synthesis. Lysenin binds to SM in the PM and forms a pore, thereby causing cytotoxicity. SPT, serine palmitoyltransferase; KDSR, 3-ketodihydrosphingosine reductase; CerS, ceramide synthase; DES, sphingolipid Δ(4)-desaturase; SMS, SM synthase; UGCG, UDP-glucose ceramide glucosyltransferase (or GlcCer synthase). **(B)** Scheme of genome-wide KO screening of lysenin-resistant genes. Two independent pools of lentivirus-based GeCKO v2 pooled libraries, each of which were composed of two-half libraries, A and B (resulting in four independent pools, A-1, A-2, B-1, and B-2), were used for the screening. A heatmap composed of sgRNAs of 93 candidate genes is shown. **(C)** Heatmaps of genes involved in the pathways of de novo SM synthesis and Golgi-PtdIns(4)P metabolism are shown. Genes identified in the screening are shown in magenta. **(B and C)** Heatmaps represent the fold-enrichment of normalized read numbers of six sgRNAs of each gene in two independent experiments. See also Table S1 and Fig. S1 for other genes. **(D)** Domains and interaction partners of PI4KB, ACBD3, and C10orf76. Numbers represent the amino acid number counted from the initial methionine (the most typical isoforms expressed in HeLa cells are shown). Regions involved in interactions with proteins or ligands are shown in the upper, colored lines. ACB, acyl-CoA binding; GOLD, Golgi dynamics; DUF, domain of unknown function.

CoA as a ligand), a glutamine-rich Q domain (which interacts with PI4KB), and a Golgi dynamic (GOLD) domain (which interacts with the Golgi membrane protein Giantin and the protein phosphatase PPM1L; Yue et al., 2019; Fig. 1 D). C10orf76 has been studied for its role in enterovirus infection (Blomen et al., 2015; McPhail et al., 2020); however, its inherent function in the host cell remains unelucidated. Recently, it has been suggested that C10orf76 binds to Golgi Brefeldin A-resistant guanine nucleotide exchange factor 1 (GBF1) and is somehow involved in Golgi maintenance and secretion (Chan et al., 2019). C10orf76 is the only gene in the human genome that encodes a DUF1741 domain, the function of which remains elusive (Fig. 1 D). Several research groups demonstrated that PI4KB and ACBD3 are pivotal host factors for the propagation of picornaviruses (Sasaki et al., 2012; Greninger et al., 2012; Lyoo et al., 2019; also reviewed in Lu et al., 2020). In addition, Burke and his colleagues showed that C10orf76 participates in the organization of the PI4KB-dependent enterovirus replication complex (McPhail et al., 2020). Nevertheless, it is unknown whether ACBD3 and C10orf76 have redundant and/or distinct physiological roles in the metabolism of sphingolipids although several interacting partners of ACBD3 and C10orf76 have been identified (Liao et al., 2019; Shinoda et al., 2012; Chan et al., 2019).

### Disruption of PI4KB, ACBD3, or C10orf76 confers lysenin resistance to HeLa cells

We constructed KO cell lines of *PI4KB*, *ACBD3*, and *C10orf76* using CRISPR/Cas9 (Fig. S2 A) and found that these KO cells acquired resistance to lysenin (Fig. 2 A). The degree of lysenin resistance was strongest in the *PI4KB* KO cells, followed by *C10orf76*, and then *ACBD3* (Fig. 2 A), which was correlated with the levels of fold-enrichment of each sgRNA during the screening (Fig. 1 C). These mutant cell lines were rescued to be lysenin-sensitive by the ectopic expression of each full-size protein fused with a short peptide tag at the N-terminus (Fig. 2 B and Fig. S2 B). Moreover, when *ACBD3* and *C10orf76* were doubly disrupted, the level of lysenin resistance of the *ACBD3/C10orf76* double KO (DKO) cells increased to the level of the *PI4KB* KO cells (Fig. 2 A). Ectopic expression of recombinant V5-ACBD3 in the *ACBD3/C10orf76* DKO cells reversed the lysenin resistance to a level similar to that of the *C10orf76* single KO cells (Fig. 2 B). These results indicated that *PI4KB* single knockout (SKO) endows full lysenin-resistance to HeLa cells but that either of *ACBD3* or *C10orf76* SKO endows only partial resistance. In addition, the additive effect of *ACBD3* and *C10orf76* KO on lysenin resistance invoked the possibility that ACBD3 and C10orf76 might have essentially distinct but partially redundant roles in the metabolism of SM.

Due to technical difficulties, we were unable to adjust the ectopic expression of HA-C10orf76 to the endogenous level of the wild-type HeLa cells, although the ectopic expression level of V5-ACBD3 was successfully adjusted to the endogenous level (Fig. S2 B). The overexpression of HA-C10orf76 in *ACBD3/C10orf76* DKO cells reversed the lysenin sensitivity to the level observed in the wild-type HeLa cells (Fig. 2 B), which suggested that when overexpressed, C10orf76 could functionally compensate for the loss of ACBD3.

### PI4KB, ACBD3, and C10orf76 are required for the CERT-mediated synthesis of SM

To examine whether PI4KB, ACBD3, and/or C10orf76 KO impact on the content of SM in cells, we determined the cellular lipidome and found that the content of SM in *PI4KB* KO and *ACBD3/C10orf76* DKO cells was ∼30% less than that of the parent cells (Fig. 2 C). In SKO cells of *ACBD3* or *C10orf76*, the SM content tended to decrease but did not change significantly compared with the SM content in the parental control (Fig. 2 C), in line with the result that these SKO cells were less lysenin-resistant than *ACBD3/C10orf76* DKO cells (Fig. 2 A). Lysenin binds to clustered SM, not mono dispersed SM molecules, in membranes (reviewed in Yilmaz et al., 2018), and only a partial decrease in the cellular SM level is enough to gain discernible resistance to lysenin in mammalian cells (Hanada et al., 1998). The content of PtdCho and other lipid types was not significantly affected in any of the mutant cell types tested, except for an increase in lactosylceramide in *ACBD3* SKO cells (Fig. 2 C and Fig. S2 C). The acyl chain profile of SM was not discernibly altered in any of the KO cell lines tested (Fig. S2 D).

We next performed metabolic labeling experiments with radioactive serine to examine whether KO of these genes affected the de novo synthesis of SM. There was decreased synthesis of SM, but not other lipid types (e.g., glucosylceramide [GlcCer], PtdSer), in these mutant cell lines: the level of SM synthesis in the PI4KB KO cells was 38.9 ± 11.5% of that of the wild-type HeLa cells (Fig. 2 D). SM synthesis in the SKO cells of *ACBD3* and *C10orf76* was 59.5 ± 8.7 and 58.3 ± 10.5% of that of the parent cells, respectively, while it was 36.0 ± 8.7% in the *ACBD3/C10orf76* DKO cells (Fig. 2 D), in agreement with the results of the lysenin resistance analysis.

Brefeldin A (BFA), an inhibitor of guanine nucleotide exchange factors (GEFs) of ADP ribosylation factor (ARF), inhibits vesicle transport between the ER and the Golgi apparatus, thereby redistributing Golgi-residing proteins into the ER (or, more precisely, an ER–Golgi merged structure; Klausner et al., 1992; Chardin and McCormick, 1999). Thus, in BFA-treated cells, ER-to-Golgi transport of ceramide by CERT is no longer required for the conversion of de novo synthesized ceramide to SM (Fukasawa et al., 1999). BFA treatment recovered the SM synthesis in all of the KO cells constructed for this study (Fig. 2 D). These findings suggested that gene disruption of *PI4KB*, *ACBD3*, or *C10orf76* results in impaired SM synthesis due to the dysfunction of CERT, not SM synthases.

To examine the suggestion further, we analyzed the intracellular movement of $C_5$-DMB-ceramide, a fluorescent analog of ceramide, which at least partially recapitulates the CERT-dependent ER-to-Golgi trafficking of natural ceramide in living cells (Fukasawa et al., 1999; Hanada et al., 2003). When cells were incubated with $C_5$-DMB-ceramide at 4°C for 30 min, intracellular reticular structures (i.e., the ER) were mainly labeled in all cell types examined (Fig. 2 E, 0 min). After chasing the prelabeled cells at 37°C for 10 min, the fluorescent signals were redistributed to the perinuclear regions in the parent cells (Fig. 2 E, see also the line profile), whereas the perinuclear redistribution of the fluorescence in the *PI4KB* SKO and *ACBD3/C10orf76* DKO cells was reduced to the similar level as in the *CERT* KO cells

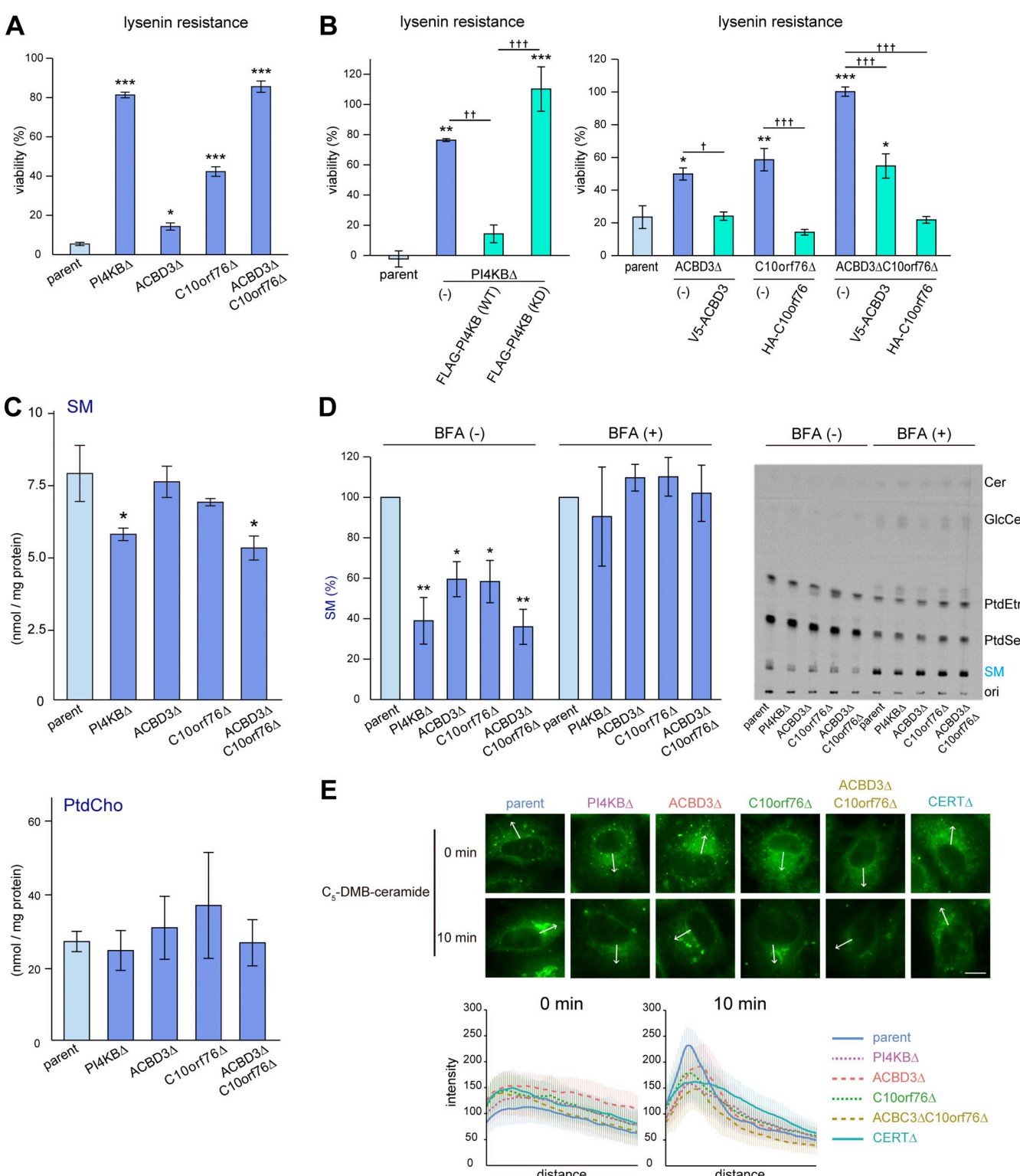

Figure 2. **Disruption of *PI4KB*, *ACBD3*, or *C10orf76* downregulates SM synthesis. (A)** Gene disruption of *PI4KB*, *ACBD3*, and *C10orf76* confers lysenin resistance to HeLa cells. **(B)** Stable expression of N-terminally tagged recombinant proteins reverses the lysenin resistance in each of the KO cell lines. **(A and B)** Overnight cultured cells were treated with lysenin at 50 (A) or 100 (B) ng/ml for 2 h. Viability was estimated via a colorimetric assay using water-soluble formazan dye and is shown as the percentage of the $A_{450\ nm}$ value in the absence of lysenin. The bar graph represents the mean ± SEM of three biological replicates. Representative data from at least two independent experiments with similar results are shown. **(C)** The content of cellular SM is decreased in *PI4KB* KO and *ACBD3/C10orf76* DKO cells. Lipids were extracted from cells subcultured in serum-free medium then analyzed by LC–MS. The bar graph represents the mean ± SEM of three independent experiments. See also Fig. S2. (See Tables S2 and S3 for the raw data sets.) **(D)** De novo SM synthesis was defective in the KO cell lines. Overnight cultured cells pretreated with mock or 1 µg/ml BFA for 30 min were metabolically labeled with L-[$^{14}$C(U)]-serine for 8 h. Lipids were extracted from cell lysates (with the equal protein amounts among the samples) and separated using TLC. The intensity of autoradiography of labeled SM was

analyzed using ImageJ, with the intensity of parent cells set to 100%. The bar graph represents the mean ± SEM of three independent experiments. A representative autoradiography image of the TLC plates is shown. **(A–D)** * and † represent statistically significant differences between the KO cells and the parent cells, or between samples linked with black lines, respectively. * and †, P < 0.05; ** and ††, P < 0.005; *** and †††, P < 0.0005. Statistical analysis of A, C, and D were performed by one-sided Dunnett test. Tukey-Kramer test was used for B. **(E)** Analysis of intracellular trafficking of $C_5$-DMB-ceramide. Cells were labeled with $C_5$-DMB-ceramide complexed with BSA for 30 min at 4°C and chased for 10 min at 37°C. Fixed cells were subjected to fluorescence microscopy observation. Graphs represent the mean ± SD of line profiles of the perinucleus regions of interest (ROIs, depicted by white arrows), calculated from n > 25 cells in at least three images. Data are representative of at least two independent experiments with similar results. Scale bar, 10 µm.

(Fig. 2 E). As expected, the impact on the perinuclear redistribution of $C_5$-DMB-ceramide was smaller in the *ACBD3* SKO and *C10orf76* SKO cells than in the DKO cells (Fig. 2 E). Collectively, these results showed that disrupting *PI4KB* and *C10orf76* (especially with co-disruption of *ACBD3*) impaired the CERT-dependent trafficking of ceramide from the ER to the Golgi site for the de novo SM synthesis.

It should also be noted that the $C_5$-DMB-ceramide-enriched regions in the *ACBD3* SKO cells did not display a ribbon-like Golgi morphology, in line with a previous study showing that *ACBD3* knockdown causes the fragmentation of the Golgi apparatus (Liao et al., 2019). However, the regions in *PI4KB* SKO, *C10orf76* SKO cells, and *ACBD3/C10orf76* DKO cells did appear to be ribbon-like (Fig. 2 E). Although we do not know how *ACBD3* SKO cells, but not *PI4KB* SKO nor *ACBD3/C10orf76* DKO cells, exhibited a strong impact on the Golgi morphology, it is possible that the unbalance in the Golgi-PtdIns(4)P distribution caused by disruption of *ACBD3*(which will be discussed later) may serve as a signal to trigger the Golgi fragmentation.

### Neither the phosphoregulation nor protein expression of CERT is compromised by disrupting PI4KB, ACBD3, and C10orf76

The function of CERT is regulated by the phosphorylation state of the serine-repeat motif (SRM); it is repressed when it receives multiple phosphorylations (Kumagai et al., 2007; Fugmann et al., 2007; Tomishige et al., 2009; Goto et al., 2022b). In SDS-PAGE, a band of CERT with a hyperphosphorylated SRM can be visually distinguished from that of CERT with non-phosphorylated or hypophosphorylated SRM because the multiple phosphorylations of the SRM affect its mobility: the upper band of the ~70-kD doublet in CERT immunoblotting images represents the hyperphosphorylated form while the lower band represents the de- or hypophosphorylated form (Kumagai et al., 2007; Fugmann et al., 2007; Tomishige et al., 2009; Goto et al., 2022b). Previous studies have suggested that devoid of cellular SM triggers dephosphorylation of the CERT SRM, which induces a conformational rearrangement of CERT and makes it more accessible to PtdIns(4)P and VAPs (Kumagai et al., 2007, 2014; Sugiki et al., 2018; Goto et al., 2022a). Thus, there was the possibility that disrupting *PI4KB*, *ACBD3*, and/or *C10orf76* upregulated the SRM phosphorylation-dependent repression of CERT. However, *C10orf76* KO rather increased the ratio of the de- or hypophosphorylated form (i.e., active form) relative to the total CERT forms (Fig. 3 A). The change in the phosphorylation state of CERT in *PI4KB* KO was modest (P = 0.07), but the total amount of CERT was significantly increased (Fig. 3 A). These results eliminated the possibility that the gene disruption compromised the production of active CERT, thereby repressing the

synthesis of SM. The results also imply that inappropriate dysfunction of CERT induces compensatory responses in the expression and phosphorylation state of the endogenous CERT in cells.

### PI4KB, ACBD3, and C10orf76 are required for Golgi recruitment of CERT

We next examined whether *PI4KB*, *ACBD3*, and *C10orf76* are involved in the distribution of CERT to ER–Golgi contacts. To this end, we observed the localization of stably expressed CERT tagged with the fluorescent protein mVenus (hereafter referred to as CERT-mVenus) in the parent and KO HeLa cells (Fig. 3 B). CERT is dispersed throughout the cell and partially enriched in the Golgi complex under normal culture conditions. The Golgi localization of CERT-mVenus was decreased in the *PI4KB* KO cells (Fig. 3, B and C), which is consistent with the previous research using knockdown cells (Capasso et al., 2017). A decrease in Golgi localization of CERT-mVenus was also observed in the *ACBD3/C10orf76* DKO cells. Note that it is possible that the morphological changes of the Golgi complex may have affected the image analysis, making it difficult to detect the difference at basal condition. When cells were treated with myriocin (also known as ISP-1), a potent inhibitor of the serine palmitoyltransferase enzyme responsible for the initial step of de novo synthesis of sphingolipids, the distribution of CERT-mVenus in the Golgi was strongly enhanced in the parent HeLa cells (Fig. 3 B), in line with the behavior of the short-peptide epitope HA-tagged CERT depicted in a previous study (Kumagai et al., 2007). The Golgi-enrichment of CERT-mVenus was decreased in the *PI4KB* KO cells under myriocin treatment compared to the parent cells, and its punctate distribution was instead observed (Fig. 3, B and C). The punctate distribution of CERT-mVenus also occurred in the *C10orf76* KO cells and was exacerbated in the *ACBD3/C10orf76* DKO cells, although we failed to detect significant changes in *ACBD3* SKO cells (Fig. 3, B and C). It is also noteworthy that CERT-mVenus puncta, observed in the *PI4KB* KO and *ACBD3/C10orf76* DKO cells, was considerably colocalized with VAP-A, indicating that the VAP-binding ability of CERT was sustained in these KO cells (Fig. 3 D). Note that the expression levels of CERT-mVenus in the KO cells were comparable to that of the parent cells (Fig. S3 A). Nonetheless, the KO of *C10orf76* still increased the ratio of the de- or hypophosphorylated form of CERT-mVenus, and the expression levels and the phosphorylation state of endogenous CERT in the KO cells tended to be unaltered by ectopically expressing CERT-mVenus (Fig. S3 A, see the contrasted image). Collectively, these results suggest that the compromised recruitment of CERT to the Golgi apparatus caused a reduction in the synthesis of SM in the KO cells.

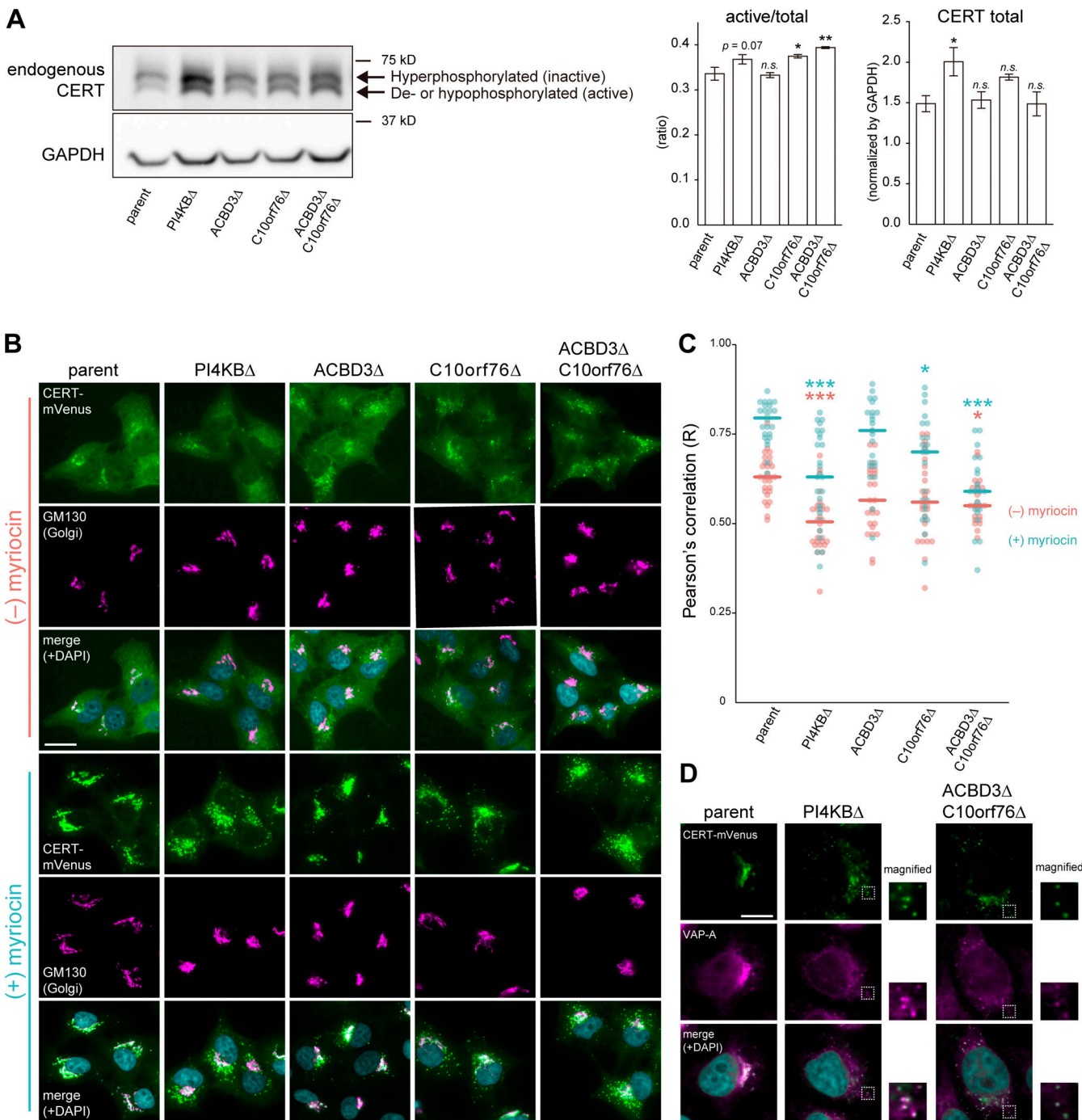

Figure 3. **Recruitment of CERT to the Golgi apparatus is impaired in *PI4KB* KO and *ACBD3/C10orf76* DKO cells. (A)** Western blot analysis of the phosphorylation state of endogenous CERT. Cell lysates were immunoblotted with anti-CERT or anti-GAPDH antibodies. CERT displayed a doublet pattern that represents a hyperphosphorylated upper band (i.e., the inactive form) and a de- or hypophosphorylated lower band (i.e., the active form). The intensities of the upper and lower bands of CERT were quantified, and the ratio was calculated by dividing the intensity of the lower bands by the sum of the upper and lower bands (left panel). To compare the relative expression levels of CERT between different cell types, the total intensity of the upper and lower bands was normalized by the intensity of GAPDH as the loading control (right panel). The graphs represent the mean ± SEM of three biological replicates. Statistical significance was determined by two-sided Dunnett test *, P < 0.05; **, P < 0.005. n.s., not significant. **(B)** Microscopic observation of the intracellular distribution of CERT-mVenus. Cells stably expressing CERT-mVenus were cultured in the presence or absence of 2.5 μM myriocin for 24 h. Fixed cells were then immunostained with an anti-GM130 antibody. **(C)** Image analysis of B. Data are representative of at least two independent experiments with similar results. The dots represent the Pearson's correlation coefficient between CERT-mVenus and the Golgi marker GM130 of one cell (*n* = 23–28), calculated from at least three images. The line segments represent the median. Orange, myriocin untreated; light blue, myriocin treated. Statistical differences, determined by Steel-Dwass test, between the parent and KO cells in each of the myriocin-treated and untreated conditions are shown. *, P < 0.05; ***, P < 0.0005. n.s., not significant. **(D)** Microscopic observation of CERT-mVenus puncta in the KO cells. Cells stably expressing CERT-mVenus were treated with 2.5 μM myriocin for 24 h. Fixed cells were then immunostained with an anti-VAP-A antibody. Magnified views of the white-dashed boxes are shown. **(B and D)** Nuclei were visualized by staining with DAPI. Scale bars, 10 μm. Source data are available for this figure: SourceData F3.

## ACBD3, C10orf76, or both, are required for the Golgi recruitment of PI4KB

In mammalian cells, the Golgi apparatus consists of at least three different cisternae: cis, medial, and trans. In addition, ER–Golgi intermediate compartments (ERGICs) and the TGN are often regarded as subregions of the Golgi complex (Pantazopoulou and Glick, 2019). Previous studies have shown that ACBD3 is involved in the recruitment of PI4KB to the Golgi complex (Sasaki et al., 2012; Klima et al., 2016) and that gene disruption of *C10orf76* causes a decrease in the PtdIns(4)P level in the Golgi complex (McPhail et al., 2020). Notably, we found that gene disruption of *ACBD3* and *C10orf76* did not affect the protein expression level of PI4KB (Fig. 4 A), although double disruption led to a modest decrease (P = 0.07). These results led us to hypothesize that ACBD3 and C10orf76 might be distinct but partially redundant determinants to recruit PI4KB to the Golgi complex.

To test this hypothesis, we observed the subcellular localization of stably expressed FLAG-PI4KB in SKO or DKO cells of *ACBD3* and *C10orf76* (Fig. 4, B and C; and Fig. S3 B). FLAG-PI4KB was distributed throughout the cytoplasm with partial enrichment at the Golgi apparatus, when expressed in *PI4KB* KO HeLa cells under normal culture conditions. The Golgi enrichment of FLAG-PI4KB was enhanced upon treatment of cells with the PI4KB inhibitor PIK93, which was in line with a previous study that used a different cell type (Blomen et al., 2015). Golgi localization of FLAG-PI4KB in the *ACBD3* SKO cells was decreased in line with previous studies, suggesting that ACBD3 is the recruiting factor for PI4KB (Sasaki et al., 2012; Klima et al., 2016). Nonetheless, PIK93-induced Golgi-enrichment of FLAG-PI4KB was still observed (Fig. 4, B and C), albeit to a lesser extent. The *C10orf76* SKO had little effect on PI4KB accumulation, whereas the Golgi localization of FLAG-PI4KB in the *ACBD3/C10orf76* DKO cells was significantly decreased (Fig. 4, B and C). These results raise the possibility that PI4KB employs at least two Golgi-recruiting mechanisms; in HeLa cells, one is dependent on ACBD3, and the other is dependent on C10orf76. Nevertheless, this possibility seemed to be inconsistent with a previous proposal that C10orf76 is recruited to the Golgi apparatus via interaction with PI4KB on the basis of the observation that artificial tethering of PI4KB to the mitochondria led to localization of C10orf76 to the mitochondria (McPhail et al., 2020). To address this discrepancy, we examined the subcellular distribution of HA-C10orf76 stably expressed in the *PI4KB* KO or *ACBD3/C10orf76* DKO cells. A portion of HA-C10orf76 was localized at the Golgi apparatus even in the absence of PI4KB or ACBD3 (Fig. S4 A), demonstrating that C10orf76 is capable of localizing at the Golgi complex in a PI4KB-independent manner under normal growth conditions. Notably, our conclusion does not exclude the possibility that PI4KB acts upstream of C10orf76 in enterovirus-infected cells. Previous studies demonstrated that viral 3A protein interacts with ACBD3, which in turn recruits PI4KB via a direct interaction between them (Greninger et al., 2013; Lyoo et al., 2019) and that PI4KB is capable of simultaneously binding to ACBD3 and C10orf76 (McPhail et al., 2020). Therefore, C10orf76 acts upstream of PI4KB to recruit PI4KB to the Golgi complex, while

PI4KB acts upstream of C10orf76 to recruit C10orf76 to enteroviral replication sites.

We next examined the intracellular distribution of PtdIns(4)P, using an anti-PtdIns(4)P monoclonal antibody to determine whether the impaired Golgi recruitment of PI4KB affected the level of Golgi-PtdIns(4)P (Fig. 4 D). PtdIns(4)P was condensed at the Golgi apparatus in the parent cells, while the *PI4KB* KO cells exhibited scarce PtdIns(4)P signals at the Golgi apparatus (Fig. 4 D). The Golgi-PtdIns(4)P level was also decreased in the *C10orf76* KO cells, as reported in a previous study (McPhail et al., 2020), whereas KO of *ACBD3* resulted in little impact on Golgi-PtdIns(4)P, in line with its reduced impact on CERT localization. The Golgi-PtdIns(4)P levels in *ACBD3/C10orf76* DKO cells were similar to those in the *PI4KB* KO cells, consistent with the defective recruitment of PI4KB in the DKO cells. While single KO of *C10orf76* had less impact on PI4KB recruitment to the Golgi apparatus (Fig. 4, B and C), its effect on Golgi-PtdIns(4)P was significant (Fig. 4 D). Golgi recruitment of PI4KB by ACBD3 may have been upregulated in the *C10orf76* SKO cells and/or C10orf76 may also affect the activity of PI4KB via GBF1–ARF1.

## ACBD3 and C10orf76 are distributed to different subregions of the Golgi apparatus

As described above, ACBD3 and C10orf76 likely have distinct but partially redundant roles in the recruitment of PI4KB to the Golgi complex. ACBD3 is known to bind the Golgi membrane protein giantin, which is localized to proximal Golgi regions (Sohda et al., 2001; Greninger et al., 2013), while SMS1, the major SMS for de novo synthesis of SM, is preferentially distributed to distal Golgi regions (Halter et al., 2007). Thus, we hypothesized that C10orf76 might be a determinant to recruit PI4KB to distal Golgi regions. When observed under confocal microscopy, ACBD3 and stably expressed HA-C10orf76 were localized at the Golgi apparatus, as previously reported (Sohda et al., 2001; Blomen et al., 2015), with substantial overlaps (Fig. 5 A). To test the hypothesis that ACBD3 and C10orf76 might be distributed to different Golgi subregions, we employed STED (stimulated emission depletion) microscopy, using a resolution level that rendered the pixel size of raw STED images equal to 20 nm. This was sufficient to clearly resolve the cis-Golgi marked by GM130 and TGN marked by TGN46 (Fig. S4 B): detectable of interspace between *cis*-Golgi and TGN, at which medial- and trans-Golgi should reside. STED microscopy observation revealed that ACBD3 and C10orf76 were predominantly contiguously distributed, or localized mutually exclusively, with only minor colocalization (Fig. 5 A, representative regions [i–iii] are magnified). Of note, HA-C10orf76 also showed punctate localization dispersed in the cytoplasm. Co-staining with a cis-Golgi marker (GM130) or a TGN marker (TGN46) also suggested that ACBD3 and HA-C10orf76 were distributed to different Golgi regions (Fig. 5 B). ACBD3 was mainly distributed adjacent to GM130 with partial overlaps, while only a small proportion of ACBD3 adjoined or overlapped with TGN46. In contrast, HA-C10orf76 was mainly localized in the vicinity of TGN46, while gaps were observed between HA-C10orf76 and GM130 (Fig. 5 B, line plots). Manders colocalization coefficient M1, calculated from the uncropped images, verified that ACBD3 colocalized

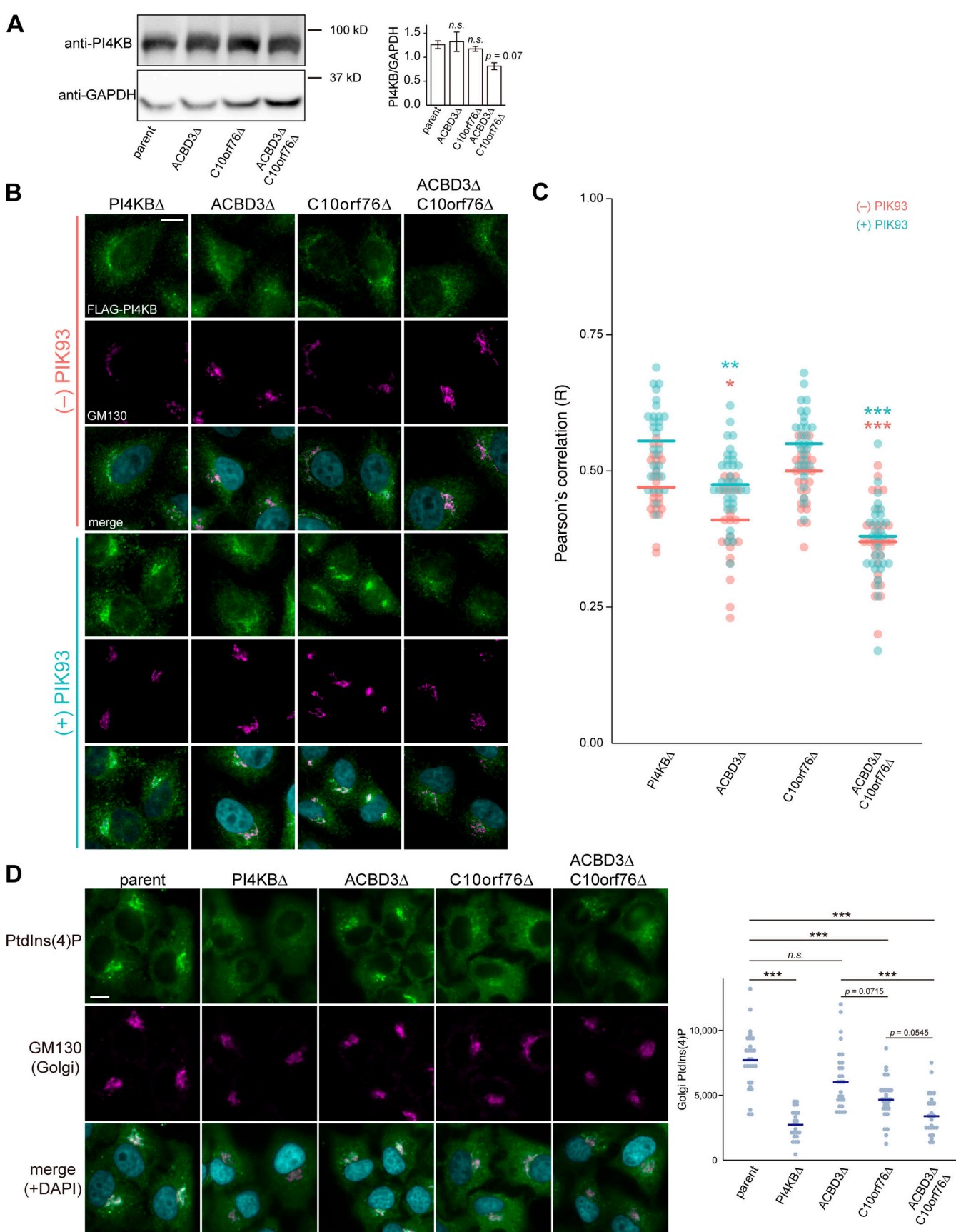

Figure 4. **PI4KB requires either ACBD3 or C10orf76 for its recruitment to the Golgi apparatus. (A)** Western blot analysis of the abundance of endogenous PI4KB. Cell lysates were immunoblotted with anti-PI4KB or anti-GAPDH antibodies. The graph represents the mean ± SEM (three biological

replicates) of the band intensity of PI4KB, which was normalized by that of the loading control GAPDH. Statistical significance was determined by two-sided Dunnett test. n.s., not significant. **(B)** Microscopic observation of FLAG-PI4KB recruitment to the Golgi apparatus. Cells stably expressing FLAG-PI4KB were treated with 0 or 2 μM PIK93 for 4 h. Fixed cells were then immunostained with anti-FLAG and anti-GM130 antibodies. **(C)** Image analysis of B. Data are representative of two independent experiments with similar results. The dots represent the Pearson's correlation coefficient between FLAG-PI4KB and the Golgi marker GM130 in one cell (n = 24–36), calculated from at least three images. The line segments represent the median. Orange, without PIK93; light blue, PIK93-treated. Statistical differences, determined by Steel-Dwass test, between the parent and KO cells in each of the PIK93-treated and untreated conditions are shown. *, P < 0.05; **, P < 0.005; ***, P < 0.0005. **(D)** Analysis of PtdIns(4)P pools in the Golgi apparatus. Fixed cells were immunostained with anti-PtdIns(4)P and anti-GM130 antibodies. The dots represent the intensity of Golgi-PtdIns(4)P normalized by the Golgi area in one cell (n = 23–30), calculated from at least three images. Data are representative of two independent experiments with similar results. The blue line segments represent the median. Statistical significance was determined by Steel-Dwass test. ***, P < 0.0005. n.s., not significant. **(B and C)** Nuclei were visualized by staining with DAPI. Scale bars, 10 μm. Source data are available for this figure: SourceData F4.

more with GM130 than TGN46, and C10orf76 colocalized more with TGN46 than GM130 (Fig. 5 B and Fig. S4 D). Furthermore, line profiles of co-stained GM130, ACBD3, and HA-C10orf76 suggested that ACBD3 was located between GM130 and HA-C10orf76 (Fig. 5 C). These results indicated that ACBD3 was preferentially distributed to proximal Golgi regions (i.e., the cis- and medial-Golgi stacks), in agreement with a previous study showing that ACBD3 is capable of associating with giantin, a membrane protein integrated at the cis- and medial-Golgi (Linstedt et al., 1995; Martínez-Menárguez et al., 2001). On the other hand, C10orf76 was predominantly distributed to distal Golgi regions (i.e., the trans-Golgi and TGN). Predictably, HA-CERT barely colocalized with GM130, while it localized adjacent to TGN46 with only partial overlap, which resembled the distribution pattern of C10orf76 (Fig. 5 D and Fig. S4 D).

STED microscopic analysis also confirmed that stably expressed SMS1-V5 was predominantly located in the distal Golgi regions (Fig. S4, C and D), in line with a previous immuno-electron microscopic study (Halter et al., 2007). Taken together, C10orf76 and CERT are likely to localize at the distal Golgi regions where the SM synthase is mainly distributed (Fig. S4 D). In addition, a portion of HA-CERT was colocalized with SMS1-V5 (Fig. 6 A). HA-C10orf76 was also partially colocalized with SMS1-V5, whereas ACBD3 was localized adjacent to SMS1-V5 with a slight overlap (Fig. 6 A). These results suggest that the distal Golgi regions, to which CERT should be recruited for the efficient metabolic flow of ceramide to SM, are likely to be pinpointed by a PtdIns(4)P pool generated by the C10orf76–PI4KB axis. Lipid molecules were shown to be discernibly mobile even after fixation by glutaraldehyde or formaldehyde (Tanaka et al., 2010), not allowing us to analyze the intracellular distribution of PtdIns(4)P at the STED super-resolution level. Of note, only a portion of HA-CERT and V5-C10orf76 significantly overlapped when observed by STED microscopy (Fig. S4 E). Moreover, there was no detectable co-immunoprecipitation of C10orf76 or PI4KB along with CERT, while PI4KB was observed to precipitate along with C10orf76 (Fig. S4 F), which is consistent with previous studies (Greninger et al., 2013; McPhail et al., 2020). Thus, it is unlikely that CERT directly interacts with C10orf76 and/or PI4KB to be recruited to the distal Golgi regions.

Moreover, STED microscopic analysis of the endogenous VAP-A stained with an anti-VAP-A antibody revealed its heterogeneous distribution within the ER (Fig. 6 B). Although the VAP-A-distribution pattern was intermittent, it could be ascribed to a part of the reticular network of the ER because VAP-

A-positive regions appeared to be connected with the KDEL-positive ER. We hypothesized that the discontinuous distribution of VAP-A might reflect certain ER MCSs because VAP-A is involved in various MCSs of the ER with other organelles (Murphy and Levine, 2016; Alli-Balogun and Levine, 2019). To test this hypothesis, we examined whether the distal-Golgi distribution pattern of CERT is related to the discontinuous distribution of VAP-A in the ER. STED microscopic analysis showed partially overlapping distributions of HA-CERT and VAP-A (Fig. 6 C, arrowheads). Notably, the distribution of HA-CERT never perfectly overlapped with that of VAP-A but was instead mostly localized in the close vicinity of VAP-A. Similarly, HA-C10orf76 was suggested to be condensed at the ER-associated or ER-adjacent Golgi subregions (Fig. 6 C). V5-ACBD3 displayed a broader distribution pattern within the Golgi stacks and was not enriched in the vicinity of VAP-A (Fig. 6 C, line plot). SMS1-V5 displayed a broader distribution than HA-CERT in the Golgi complex (Fig. 6 A) and was not enriched in the vicinity of VAP-A (Fig. 6 C), ruling out the possibility that proteins embedded in the distal Golgi were non-specifically enriched in the vicinity of VAP-A. Collectively, these results suggested that a considerable portion, if not all, of the Golgi-attached CERT was selectively localized to the ER–distal Golgi MCSs.

## Discussion

LTP-mediated inter-organelle transport of lipids at MCSs is critical not only for the metabolism of lipids but also for their functions in cells (Hanada, 2018; Balla et al., 2019; Wong et al., 2019). Although it is widely accepted that PtdIns(4)P plays a key role in recruiting specific LTPs to the Golgi apparatus (Tóth et al., 2006; Balla et al., 2005), the precise mechanism for generating PtdIns(4)P pools at distinct MCSs has remained unclear. Here, we found that ACBD3 and C10orf76 act as mediators to recruit PI4KB to different Golgi regions. Additionally, the C10orf76–PI4KB axis was found to play a role in producing PtdIns(4)P for the recruitment of CERT to the distal Golgi regions. Once there, CERT facilitates the transfer of ceramide from the ER for the production of SM. Metabolic labeling of lipids with radioactive serine and intracellular movement of $C_5$-DMB-ceramide provided compelling evidence that disrupting the C10orf76–PI4KB axis compromises both de novo synthesis of SM and ER-to-Golgi transport of ceramide in living cells (Fig. 2, D and E), that are likely due to the dysfunction of CERT. Clear impacts of the C10orf76 disruption on the Golgi-localization of

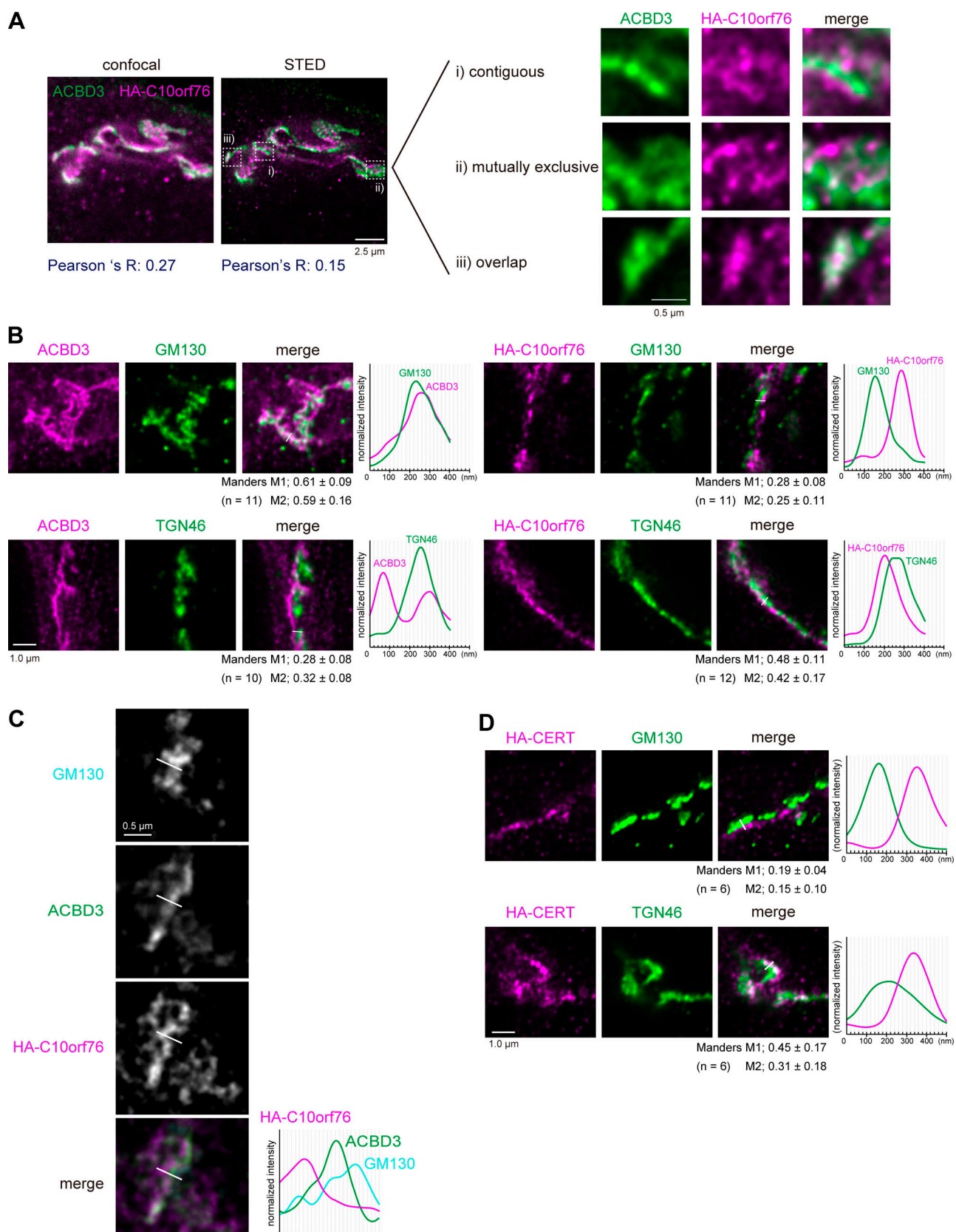

Figure 5. **ACBD3 and C10orf76 localize at distinct subregions in the Golgi apparatus. (A)** STED microscopy observation of ACBD3 and HA-C10orf76. The *C10orf76* KO cells stably expressing HA-C10orf76 were immunostained with anti-ACBD3 and anti-HA antibodies and imaged using confocal or super-resolution STED microscopy. **(B)** STED microscopic observation of the intra-Golgi distribution patterns of ACBD3 and C10orf76. ACBD3 and HA-C10orf76 were co-stained

with a cis-Golgi marker, GM130, or a TGN marker, TGN46. **(C)** GM130, ACBD3, and HA-C10orf76 were co-stained and observed under STED microscopy. **(D)** STED microscopic observation of the intra-Golgi distribution patterns of HA-CERT. The *CERT* KO cells stably expressing HA-CERT were immunostained with anti-HA + anti-GM130 antibodies or anti-HA + anti-TGN46 antibodies and imaged using STED microscopy. **(B–D)** Graphs represent the line profiles of ROIs (depicted by white lines): x-axis, distance; y-axis, normalized signal intensities. Scale bars, 1.0 µm. **(B and D)** The mean ± SD of Manders correlation coefficient (MCC) and the cell numbers (*n*) analyzed are shown. MCC M1 represents the colocalization rate of the protein of interest with the marker protein. M2 represents the colocalization rate of the marker protein with the protein of interest. Source data are available for this figure: SourceData F5.

CERT were observed upon myriocin treatment (Fig. 3, B and C), although the Pearson's correlation values of CERT-mVenus and the Golgi marker only marginally differed between parental cells and *C10orf76* KO cells in the absence of myriocin. This may be at least partially accounted for by a compensatory increase of the active form of CERT-mVenus in the *C10orf76* KO cells (Fig. S3 A): since the difference in the conformation of CERT is likely to affect the subcellular distribution. Myriocin treatment increased the dephosphorylated form of CERT to almost the same extent in all five cell lines examined (Fig. S3 A), allowing us to examine the effects of the gene disruptions on CERT localization under the conditions where gene disruption-dependent alterations of the phosphorylation states of CERT-mVenus were minimized (Fig. 3 C). Collectively, these results indicated that the C10orf76-PI4KB axis contributes to the recruitment of CERT to the Golgi complex. Acquiring the lysenin resistance by disrupting the C10orf76–PI4KB axis via compromising the function of CERT was in line with previous studies showing that the dysfunction of CERT endows lysenin-resistance to cells (Hanada et al., 2003; Yamaji and Hanada, 2014; Murakami et al., 2021). However, we do not deny the possibility that disrupting the C10orf76–PI4KB axis might also have compromised the transport of SM from the distal Golgi to the PM in the present study.

To date, ACBD3 has been the focus for playing the lead role in the Golgi localization of PI4KB. However, our data revealed that C10orf76 is also involved in the recruitment of PI4KB to the Golgi complex (Fig. 4 A). The STED analysis revealed that the intra-Golgi distribution patterns of ACBD3 and C10orf76 were discernibly distinct although they partly overlap (Figs. 5 and 6). ACBD3 was distributed broadly in the proximal Golgi regions, whereas C10orf76 was primarily distributed to a limited region within the distal Golgi regions, in close proximity to VAP-A positive ER subregions. Together, these findings suggest that ACBD3 mainly recruits PI4KB to the proximal Golgi and to a lesser extent to the distal Golgi, while C10orf76 is a key factor in the recruitment of PI4KB to the distal Golgi regions, which likely form the ER-distal Golgi contact zones (Fig. 7). Previous studies showed that intra-Golgi distribution patterns of PtdIns(4)P is gradient, not homogenous (D'Angelo et al., 2008; Del Bel and Brill, 2018). PtdIns(4)P levels at the proximal Golgi are kept low due to degradation by a lipid phosphatase, SAC1, which shuttles between the ER and proximal Golgi (Cheong et al., 2010; Del Bel and Brill, 2018). Our results are in line with this concept. Disruption of *ACBD3* partially impaired PI4KB recruitment to the Golgi apparatus, whereas it had little impact on the Golgi–PtdIns(4)P level. Conversely, single KO of *C10orf76* apparently did not decrease the Golgi-localizing PI4KB although it did reduce the PtdIns(4)P level in the Golgi complex (Fig. 4, B–D). In the *C10orf76* single KO cells, PtdIns(4)P production via the

ACBD3–PI4KB axis might be upregulated via increasing Golgi localization of PI4KB to compensate for the shortage of distal Golgi-PtdIns(4)P due to the absence of the C10orf76–PI4KB axis. Partial functional compensation of ACBD3 and C10orf76 to produce PtdIns(4)P at the distal Golgi regions may explain why the disruption of both *ACBD3* and *C10orf76* was required to reach the levels of the lysenin-tolerant and SM-deficit that were attained by the *PI4KB* single disruption (Fig. 2). At present, it remains unclear how CERT preferentially binds to the C10orf76-dependent PtdIns(4)P pool. Interestingly, the PH domains of CERT, OSBP, and FAPP1 have structural similarity and also common biochemical activity, in that among the various phosphoinositides, they preferentially bind to PtdIns(4)P. Nevertheless, the PH domains of OSBP and FAPP1, but not that of CERT, require ARF1 to be recruited to the Golgi complex (Levine and Munro, 2002); see also the evidence for the structural biology of the PtdIns(4)P-preferential PH domains (Sugiki et al., 2012; Prashek et al., 2013; He et al., 2011; Liu et al., 2014). Of note, the co-immunoprecipitation assay suggested that C10orf76 or PI4KB did not directly interact with CERT (Fig. S4 F). In addition, no physical interactions between CERT (also known as CO-L4A3BP) and PI4KB or C10orf76 were found in a human protein-protein interaction database (http://www.interactome-atlas.org/assessed on August 17, 2022). STED microscopy observation also revealed that only a portion of CERT colocalized with C10orf76 (Fig. S4 E). Nonetheless, there remains the possibility that a yet-to-be-identified accessory protein(s) is required for CERT to be recruited to the distal Golgi in a C10orf76–PI4KB axis-dependent manner.

SMS1, which serves as the primary SMS for de novo synthesis of SM in the Golgi complex (Tafesse et al., 2006; Tachida et al., 2020), is distributed more to the distal Golgi regions than the proximal regions (Fig. S4, C and D), while the activity of UGCG is higher in the proximal regions (Allan and Obradors, 1999; Halter et al., 2007; Yamaji and Hanada, 2015; Hayashi et al., 2018; Ishibashi et al., 2018). Mysteriously, although SMS and the GlcCer synthase UGCG share ceramide as their substrate in the Golgi complex, the de novo synthesis of SM, but not GlcCer, depends on CERT (Hanada et al., 2003; Yamaji et al., 2016). Two scenarios have been proposed to explain this. One scenario is that the GlcCer synthase activity is inhibited by PtdIns(4)P (Ishibashi et al., 2018), thereby repressing the function of UGCG at the CERT-recruited ER–Golgi contact zones, in which PtdIns(4)P is thought to be enriched. The second scenario is that the association of UGCG with SMS1 represses the UGCG activity, thereby inactivating UGCG when SMS1 is colocalized in the same compartment (Hayashi et al., 2018). Based on the results of our present study, we propose a third scenario: the preferential recruitment of CERT to the C10orf76–PI4KB axis-dependent

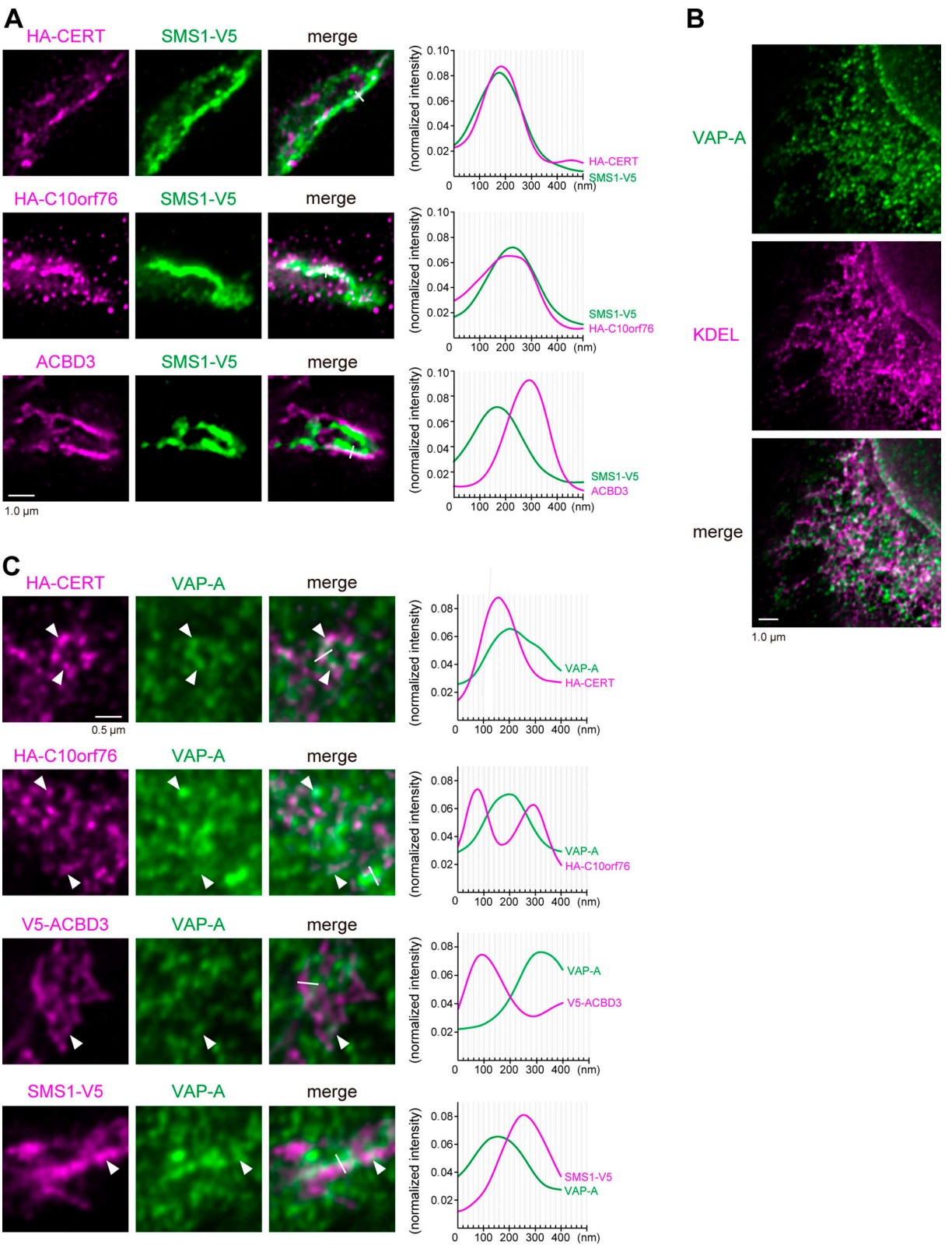

Figure 6. **C10orf76 marks the ER–Golgi ceramide transport zones. (A)** STED microscopic observation of the SMS1-positive Golgi stacks. The *CERT* KO cells stably expressing HA-CERT and SMS1-V5 or *C10orf76* KO cells stably expressing HA-C10orf76 and SMS1-V5 were fixed and immunostained with anti-ACBD3 + anti-V5 antibodies or anti-HA + anti-V5 antibodies and imaged using STED microscopy. Scale bar, 1.0 μm. **(B)** STED microscopic observation of the intra-ER distribution pattern of VAP-A. HeLa cells were fixed and immunostained with anti-KDEL and anti-VAP-A antibodies and imaged using STED microscopy. Scale bar, 1.0 μm. **(C)** STED microscopic observation of ER–Golgi-associated subregions. Cells were fixed and immunostained with anti-HA + anti-VAP-A antibodies or anti-V5 + anti-VAP-A antibodies and imaged using STED microscopy. Arrowheads indicate the overlapping regions. Scale bar, 0.5 μm. **(A and C)** Graphs represent the line profiles of ROIs (depicted by white lines): x-axis, distance; y-axis, normalized signal intensities. Source data are available for this figure: SourceData F6.

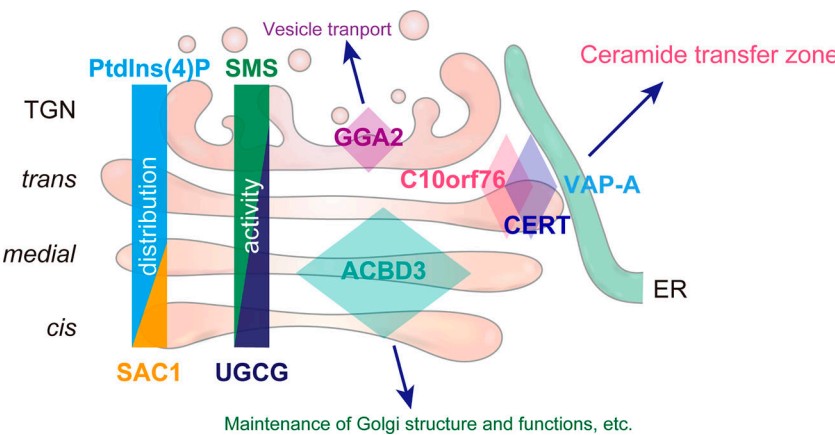

Figure 7. **The C10orf76–PI4KB axis orchestrates the CERT-mediated ceramide trafficking to the distal Golgi.** PtdIns(4)P is distributed with a gradient in the Golgi complex, being higher at the distal side. PI4KB properly adopts several mechanisms to localize at distinct Golgi regions: ACBD3 is for localizing at the cis- and medial-cisternae, C10orf76 is for localizing at the limited regions within the trans-cisterna, and GGA2 is for localizing at the TGN. PtdIns(4)P pools generated at these sites should harbor distinctive properties. In addition, PI4K2A-dependent PtdIns(4)P pools are likely to be present at the TGN. Among the diverse PtdIns(4)P pools in the Golgi complex, CERT preferentially utilizes the PtdIns(4)P pools generated by PI4KB recruited by C10orf76. This system may act as an inter-organelle metabolic channeling mechanism for the efficient conversion of ceramide to SM. Under C10orf76-deficient conditions, ACBD3-dependent PtdIns(4)P may partially compensate for the loss of C10orf76-dependent PtdIns(4)P. As ACBD3 is known to interact with a variety of proteins, ACBD3-dependent PtdIns(4)P pools may act as a scaffold for several zones that are involved in the maintenance of Golgi structure and functions. GGA2, as a clathrin adapter, likely produces PtdIns(4)P pools critical for vesicle transport.

PtdIns(4)P pool means it acts specifically at a limited subregion in the Golgi complex, to which SMS1, but not UGCG, is primarily localized (Fig. 7). This could be a simple but rational system to accomplish the rapid and accurate delivery of ceramide from the ER to the site of SM synthesis without consuming the common substrate ceramide to GlcCer in the proximal Golgi regions (Fig. 7). This model might represent one of the mechanisms underlying metabolic channeling events (Chen and Silver, 2012; Pareek et al., 2021), although all of the above-described three scenarios may operate in cells.

Super-resolution imaging revealed that colocalization of CERT and SMS1-V5 occurs in limited regions under normal growth conditions (Fig. 6 A). SMS1 was observed to be broadly distributed to the distal Golgi regions, including the trans-Golgi stacks and the TGN (Fig. S3 C), in line with the findings of a previous study (Halter et al., 2007). In contrast, CERT was largely distributed to specific zones in the distal Golgi stacks, which associate with or are adjacent to VAP-A positive ER (the zones most likely represent ER–Golgi contact sites; Fig. 5 D and Fig. 6 C). These results suggest the existence of functional zones of CERT-mediated SM synthesis (i.e., the major pathway) at the ER–Golgi contact sites. In addition, less-CERT-dependent (or independent) zones may exist in the TGN. Accordingly, it should also be noted that although GGA2 has been suggested to be involved in PI4KB recruitment to the TGN (Daboussi et al., 2017), this was not observed in our screening (Fig. S1). Considering that GGA2 acts as an Arf-binding clathrin adapter in vesicular membrane trafficking events (Hirst et al., 2000; Bonifacino, 2004), the absence of GGA2 in our screening result might be accounted for by the notion that GGA2-dependent PI4KB-recruitment occurs in the TGN or the post-Golgi region, to which CERT is rarely recruited.

PtdIns(4)P is one of the critical factors that determine the functional zones. The diversity of PtdIns(4)P molecular species (i.e., acyl chain profile) and the degree of polar-headgroup density in the membrane may confer distinct properties on the local membrane environment. PtdIns 4-kinases and their binding proteins also likely impact on the profiles of the zones. For instance, PI4K2A is known to target the membrane via its palmitoylation in a cholesterol-dependent manner (Lu et al., 2012), which may lead to the formation of PtdIns(4)P pools within a cholesterol-rich environment. Proteins segregated or recruited to the PtdIns(4)P-rich microdomains are also likely to affect the local environment. Further studies on the recruiting mechanism of C10orf76 to the distal Golgi would help us to clarify the hallmark of the PtdIns(4)P pools that CERT preferentially utilizes as a ceramide transfer zone. It is of interest to determine whether C10orf76 recognizes the ER-associated distal Golgi regions or if the recruited C10orf76 induces the contact between the distal Golgi and the ER.

C10orf76 is a typical name to refer to hypothetical or predicted genes that have not yet been characterized although it has been predicted to contain an Armadillo repeat structure. Since this study demonstrated that C10orf76 acts as a mediator to recruit PI4KB to the distal Golgi regions, we would like to propose DGARM as an alternative name for C10orf76 after a distal Golgi Armadillo repeat protein.

## Materials and methods

### HeLa cell lines and cell culture

The HeLa-mCAT #8 clone, a derivative of the HeLa ATCC CCL-2, which stably expresses the mouse ecotropic retroviral receptor mCAT1 (Yamaji et al., 2010), was used as the parental cell line in this study. The HeLa CERT KO #14 clone was established as described previously (Yamaji and Hanada, 2014). The HeLa CERT KO/HA-CERT (Goto et al., 2022a) was used as a cell line ectopically expressing HA-tagged CERT in CERT KO cells.

DMEM High Glucose (044-29765; Fujifilm Wako Pure Chemicals Corporation) supplemented with 10% FBS (172012; Sigma-Aldrich, or FBS-12A, Capricorn Scientific) and penicillin–streptomycin (P4333; Sigma-Aldrich) was used for cell culture, unless otherwise mentioned. HeLa cells were maintained in the culture medium in a 5% $CO_2$ atmosphere, 100% humidity at 37°C.

### Reagents and antibodies

Lysenin (Lot. M0217L) was a gift from Dr. Y. Sekizawa (Zenyaku Kogyo Co., Tokyo, Japan).

Other chemical reagents used in this study are listed in Table S4. The information of the antibodies used in this study is listed in Table S5.

## Genome-wide CRISPR KO screening for lysenin resistance

HeLa-Cas9/GeCKO v2 library cells (Yamaji et al., 2019) were seeded in nine 150-mm dishes with $5 \times 10^6$ cells in each and cultured overnight; six were for lysenin treatment and three were for the non-treatment control. For lysenin treatment samples, cells were treated with 100 ng/ml lysenin in DMEM for 2 h. Note that lysenin treatment was carried out under serum-free conditions. For culturing, the medium was then changed to DMEM supplemented with 10% FBS. Proliferated cells were re-seeded to avoid local cell crowding. The second and third lysenin treatments were carried out in the same way as the first treatment. After three rounds of lysenin treatment and following proliferation, genomic DNA was purified from the cells. For the non-treatment control, cells were subcultured throughout the period of the lysenin treatment, and genomic DNA was purified from the cells. DNA regions encoding sgRNAs were specifically amplified from the genome and subjected to next-generation sequencing (NGS) analysis. The preparation of NGS samples was carried out as previously described (Yamaji et al., 2019). PhiX Control Kit v3 (Illumina) was added to the samples at ~20% concentration, and Miniseq Reagent Kit (Illumina) was used for Miniseq (Illumina) sequencing. For NGS data processing and analysis, total raw read sequences were divided into each sample using an in-house program (Data S1 CRISPR_process.pl) to perform demultiplexing of fastq sequence data.

Normalization was performed as follows; normalized reads per sgRNA = reads per sgRNA/total reads for all sgRNAs in sample $\times 10^7$.

Fold enrichment was calculated as follows; Fold enrichment = normalized reads in lysenin-treated sample/normalized reads in untreated sample.

When the normalized reads in the untreated sample was 0, fold enrichment was calculated by formatting 0 to 1. Raw data and processed data of NGS are deposited to National Center for Biotechnology Information GEO (accession no. GSE185404). The sgRNAs reproducibly representing more than fivefold enrichment in both cell libraries (A and B) were selected as lysenin-resistance sgRNA candidates. Heatmaps were created using the geom tile function of ggplot2.

## Construction of knockout cell lines using CRISPR/Cas9

For the construction of CRISPR plasmids, a 20-mer guide sequence was inserted in the BsmBI-site of the pSELECT-CRISPR-CAS9 plasmid (Ogawa et al., 2018), which enables the removal of untransfected cells by puromycin treatment. The DNA sequences encoding sgRNAs used in this study are listed in Table S6.

Next, $1.5 \times 10^5$ cells were seeded in 12-well plates and cultured overnight. CRISPR plasmid (1.0 µg) was transfected to the cells using X-tremeGENE HP (6366244001; Roche). Transfected cells were transferred to 6-well plates the next day and cultured with medium containing 5 µg/ml puromycin for 3 d to exclude untransfected cells. Cells were then subcultured in puromycin-free

medium for 3–4 d. Gene-edited clones were obtained by limiting dilution, and the isolated clones were subjected to indel analysis, as previously described (Yamaji and Hanada, 2014). Gene disruption was also confirmed by Western blotting using specific antibodies (Fig. S2).

## Retroviral expression of recombinant proteins

For construction of the retroviral expression plasmids, open reading frames (ORFs) were inserted into pMXs-IRES-Puro (pMXs-IP) or pMXs-IRES-Blasticidin (pMXs-IB) retroviral vectors (Cell Biolabs Inc.). The pMXs-IB_CERT-mVenus plasmid was constructed by Dr. N. Tamura (National Institute of Infectious Diseases, Tokyo, Japan). Retroviruses were produced in Plat-E cells (RRID:CVCL_B488; Morita et al., 2000). To select retrovirus-infected cells, puromycin, blasticidin-S, or G418 was used, at concentrations of 2, 7.5, and 400 µg/ml, respectively.

For construction of the plasmids encoding FLAG-PI4KB, a DNA fragment of the coding sequence (CDS) of N-terminally FLAG-tagged PI4KB were amplified by PCR and cloned into the XhoI-NotI site of pMXs-IP. The kinase dead mutation (D656A) was introduced by PCR. The plasmid encoding N-FLAG-PI4KB for PCR templates was a gift from Dr. M. Arita (Arita, 2019, National Institute of Infectious Diseases, Tokyo, Japan).

Human ACBD3 CDS was amplified by PCR from HeLa mCAT #8 cDNA and inserted into the BamHI-NotI site of pMXs-IB, together with a V5 tag at the N-terminus. Note that a natural variant with an E187D substitution (NM_022735.4:c.561A>C) was expressed in HeLa mCAT #8 and was used in this study.

Human C10orf76 CDS (NP_078817.2, codon-optimized) was synthesized by Eurofins Genomics and cloned into the BamHI-NotI site of pMXs-IB or pMXs-IP, together with an HA or V5 tag at the N-terminus, respectively.

Primers used for CDS amplification or base substitution are listed in Table S7. Oligo DNA pairs were annealed to produce dsDNA fragments for N- or C-terminal tagging. The sequences of oligo DNA are listed in Table S8. Plasmids used in this study are listed in Table S9.

## Lysenin treatment and cell viability assay

A total of $2.5 \times 10^4$ cells was seeded in 12-well plates and cultured overnight. Cells were rinsed with DMEM, then treated with DMEM containing lysenin at 50 or 100 ng/ml for 2 h at 37°C. The medium was changed to DMEM containing Cell Counting Reagent SF (07553-44; Nacalai Tesque) and incubated for 1 h at 37°C. Cell viability was then analyzed according to the manufacturer's instructions and is shown as the percentage of the $A_{450 \text{ nm}}$ value in the absence of lysenin. Note that lysenin treatment was carried in FBS-free medium and at a low cell density to avoid cell crowding.

## Cellular lipidome analysis

To avoid the introduction of serum-derived lipids, cells were subcultured in Opti-MEM (31985-070; Thermo Fisher Scientific) before the main culturing for lipid extraction. Specifically, $1.0 \times 10^6$ cells were cultured in a 10-cm dish with Opti-MEM until cells were ~90% confluent. Cells were detached from the dish using 500 µl of cell dissociation buffer (13151-014; Thermo Fisher

Scientific), and $1.0 \times 10^6$ cells were passaged on a 10-cm dish with Opti-MEM until they were ~90% confluent. Similarly, $1.0–1.5 \times 10^6$ cells were passaged again on a 10-cm dish and cultured until they were ~80% confluent. Cells were washed twice with ice-cold PBS, then scraped off using 800 µl of ice-cold KCl–HCl solution (0.9% KCl, 25 mM HCl). Cells were disrupted using a probe-type sonicator to obtain total cell lysate, and supernatants obtained following centrifugation at 1,000 rpm for 3 min at 4°C were used for a BCA protein assay (23227; Thermo Fisher Scientific). Lipid samples for mass spectrometry (MS) analysis were prepared as previously described (Tachida et al., 2020). Lipids were extracted from total cell lysates supplemented with 1 nmol of each internal standard. They were then divided into two halves; one half was subjected to mild-alkali degradation for sphingolipid analysis. Internal standards (C17 SM, C17 Cer, C17 LacCer, C17 GlcCer, C17 Gb3, C15:0-18:1(d7)-PtdCho, C15:0-18:1(d7)-PtdEtn, C16:0(d31)-18:1-PtdSer, and C15:0-18:1(d7)-PtdIns) were purchased from Avanti Polar Lipids. Lipids were dissolved in 150 µl of acetonitrile/methanol (50/50, vol/vol) and analyzed using a liquid chromatography–mass spectrometry (LC–MS) system, as previously described (Nakao et al., 2019), with minor changes: a ZIC-HILIC column (5.0 µm, 2.1 × 150 mm; Merck) was used for the phospholipid analysis, and data analysis was performed using SCIEX-OS software (SCIEX; see Tables S2 and S3 for the raw data sets).

## Metabolic labeling of lipids

A total of $2 \times 10^5$ cells was seeded in a 6-well plate and cultured overnight. The medium was changed to DMEM supplemented with Nutridoma-SP (11011375001; Roche) and preincubated with mock or 1.0 µg/ml BFA for 30 min. Then, 250 nCi/well of L-[$^{14}$C(U)]-serine (MC265; Moravek) was added to the medium and incubated for 8 h. Cells were washed twice with ice-cold PBS, lysed with ice-cold 0.1% SDS solution, and the lysates were then adjusted to equalize the protein concentration within each experiment. Lipids were extracted from the lysate using Bligh and Dyer's method (Bligh and Dyer, 1959) and separated by TLC, as previously described (Yamaji and Hanada, 2014). The developing solvent was composed of methyl acetate/n-propanol/chloroform/methanol/0.25% KCl (50/50/50/20/18). Radioactive lipids on the TLC plates were detected using a BAS imaging plate (BAS IP MS 2040, 28956474; Fujifilm) and Typhoon FLA7000 (GE Healthcare). The signal intensities of the autoradiograph were analyzed using ImageJ software (National Institutes of Health).

## Analysis of C$_5$-DMB-ceramide trafficking

Cells were cultured on coverslips (Matsunami Glass) prelabeled with C$_5$-DMB-ceramide as described previously (Fukasawa et al., 1999). Briefly, cells were incubated with 1.25 µM C$_5$-DMB-ceramide complexed with 1.25 µM fatty acid-free BSA at 4°C for 30 min. Cells were washed three times with DMEM and subsequently chased in DMEM, supplemented with Nutridoma-SP, containing 0.34 mg/ml fatty acid-free BSA at 37°C for 10 min. After washing with ice-cold PBS, cells were fixed with 2% formaldehyde (methanol-free, 28906; Thermo Fisher Scientific) + 0.125% glutaraldehyde (17003-92; nacalai tesque)/PBS for

10 min at room temperature (RT). Samples were washed three times with 100 mM NH$_4$Cl/PBS and then rinsed with PBS. Microscopy observation was carried out soon after fixation.

## Immunofluorescence staining

Cells were cultured on coverslips (Matsunami Glass) and fixed with Mildform 10N (133-10311; Fujifilm Wako Pure Chemicals Corporation) at RT. The following steps were all conducted at RT unless stated otherwise. Following rinsing with PBS and incubation with 100 mM NH$_4$Cl/PBS, cells were permeabilized with 0.1% Triton X-100/PBS. Rinsed cells were incubated with 3% BSA/PBS blocking buffer. Cells were incubated with primary antibody solution. Rinsed cells were incubated in the dark with fluorophore-conjugated (Alexa Fluor 488 or Alexa Fluor 594) secondary antibody solution containing DAPI (1 µg/ml). Each antibody was diluted in 1% BSA/PBS. After washing with PBS, coverslips were immobilized on glass slides (Matsunami Glass) using Fluoromount (K 024; Diagnostic Biosystems). Upon use of anti-FLAG antibody, Triton-X100 was added to the blocking buffer and antibody solution at a concentration of 0.3%, and primary antibody-incubation was carried out overnight at 4°C.

## PtdIns(4)P analysis

Golgi-localizing PtdIns(4)P was immunostained with anti-PtdIns(4)P antibody, as previously described (Hammond et al., 2009), with some modifications. The following steps were conducted at RT, unless stated otherwise. Cells were cultured in µ-Slide 8-well chambers (ib80826; ibidi) and fixed with 2% formaldehyde (methanol-free, 28906; Thermo Fisher Scientific)/PBS for 15 min. Cells were rinsed three times with 50 mM NH$_4$Cl/PBS and permeabilized with 20 µM digitonin (D-141; Sigma-Aldrich)/buffer A (20 mM PIPES (pH 6.8), 137 mM NaCl, 2.7 mM KCl) for 5 min. After rinsing three times with buffer A, cells were incubated with blocking buffer (5% normal goat serum, 005-000-001; Jackson ImmunoResearch; 50 mM NH$_4$Cl/buffer A) for 45 min. Anti-PtdIns(4)P antibody solution was prepared at 1/300 dilution in 5% normal goat serum/buffer A. Cells were treated with anti-PtdIns(4)P antibody solution at 4°C overnight. Cells were rinsed three times with buffer A then treated with fluorophore-conjugated (Alexa Fluor 488 or Alexa Fluor 594) secondary antibody solution for 45 min. Cells were rinsed three times with buffer A and once with distilled water. Cells were covered with mounting solution (Fluoromount) and then observed under a fluorescence microscope. Golgi-PtdIns(4)P was calculated as PtdIns(4)P signal intensity within the Golgi regions. The Golgi region was selected from the binary data of GM130 images.

## Immunofluorescence staining for STED microscopy observation

Cells were fixed and immunostained as previously described (Conrad et al., 1989), with minor modifications. In brief, 10% formalin was prepared by diluting formalin (16061-00; Kanto Chemical) with a cytoskeleton stabilizing buffer (CSB; 5 mM PIPES [pH 6.1], 137 mM NaCl, 5 mM KCl, 1.1 mM Na2HPO4, 0.4 mM KH2PO4, 4 mM NaHCO3, 2 mM MgCl2, 5.5 mM glucose, 2 mM EGTA). Cells were cultured on coverslips (Matsunami, No. 1S) for 2 d and fixed with 10% formalin supplemented with 0.1%

glutaraldehyde (G-5882; Sigma-Aldrich) for 10 min at RT. (For staining with an anti-VAP-A antibody, 10% formalin was used for fixation.) The following steps were also conducted at RT. Cells were rinsed with CSB and permeabilized with 0.5% Triton X-100/CSB for 90 s. Rinsed cells were incubated with 50 mM glycine/CSB for 20 min, followed by 5.0 mg/ml NaBH$_4$/CSB for 20 min. Cells were blocked with 3% BSA/CSB for 30 min, then rinsed with CSB. Cells were incubated with primary antibody solution. Rinsed cells were then incubated with fluorophore-conjugated secondary antibody solution in the dark. After washing with CSB, coverslips were mounted on glass slides with ProLong Diamond Antifade Mountant (P36965; Thermo Fisher Scientific). Alexa Fluor 488 and Alexa Fluor 555 or DyLight 550 dyes were used for double staining, and Alexa Fluor 488, Alexa Fluor 532, and TRITC dyes were used for triple staining.

### Microscopic observation
Fluorescence microscopy images were obtained using a BZ-X 710 microscope (Keyence) equipped with a Plan Apo VC 60× Oil, NA 1.40, WD 0.13 objective (Nikon). BZ-X (Keyence) DAPI (OP-87762), GFP (OP-87763), TRITC (OP-66873), and YFP (OP-66840) were used as fluorescence filters.

STED microscopy and confocal microscopy observations were obtained as previously described (Hagihara et al., 2021), using a TCS SP8 STED 3X systems microscope (Leica Microsystems). It was equipped with a 660-nm laser for the STED donut beam, and a white laser was used for excitation. Raw STED images (pixel size 20 nm) were recorded using an HCPL APO CS2 100×/1,40 oil immersion objective (Leica Microsystems) and deconvolved with Huygens software (Scientific Volume Imaging).

### Image analysis
Microscopic images were analyzed using Fiji (ImageJ). Background subtraction was carried out equally for all the images in the same experiment. For the Golgi-recruitment analysis, cells were manually outlined for the selection of cellular regions, and the Pearson's correlation coefficients between the protein of interest and GM130 within the cellular regions were calculated by the coloc2 ImageJ plugin. To analyze the STED data, Pearson's correlation coefficient and Manders colocalization coefficient were calculated from the uncropped Golgi images (see the Source Data files) using the GDSC ImageJ plugin.

### Immunoprecipitation
Immunoprecipitation was performed as previously described (Shimasaki et al., 2022). Briefly, equal amounts of cellular lysates were incubated with anti-HA antibody-conjugated agarose (A2095; Merck Millipore) for 1 h at 4°C. The agarose beads were washed twice and incubated with 1× NuPAGE lithium dodecyl sulfate sample buffer (NP008; Invitrogen) supplemented with 50 mM dithiothreitol at 70°C for 5 min. After snap centrifugation, the supernatants were subjected to immunoblotting.

### Statistical analysis
A one-sided Dunnett's test for Fig. 2, A, C, and D and two-sided Dunnett's test for Fig. 3 A, Fig. 4 A, Fig. S2 C; and Fig. S3, A and B

were applied to test the statistical significance of any differences between KO cell lines and the parent cell line. A one-sided Dunnett's test was applied for Fig. S4 D to test the statistical significance of any difference between the marker protein and the proteins of interest. The Tukey–Kramer test was applied for statistical tests for assays using cells expressing recombinant proteins (Fig. 2 B). The Steel–Dwass test was applied for statistical tests for image analysis (Fig. 3 C, Fig. 4, C and D; and Fig. S4 A). All statistical analyses were executed in R, and the R packages multcomp and NSM3 were used for the Dunnett's and Steel–Dwass tests, respectively.

### Online supplemental material
Fig. S1 is related to Fig. 1. Table S1 contains the full data of the CRISPR screening. The raw and processed data of NGS have been deposited to National Center for Biotechnology Information GEO (accession no. GSE185404). Data S1 was used for NGS data processing and analysis. Fig. S2 is related to Fig. 2. The raw data of lipidomic analysis are provided in Tables S2 and S3. Fig. S3 A is related to Fig. 3, B and C. Fig. S3 B is related to Fig. 4, B and C. Fig. S4 A shows the subcellular distribution of HA-C10orf76 in the absence of PI4KB or ACBD3. Fig. S4, B and C show the STED microscopic images of the Golgi marker proteins and V5-SMS1. Fig. S4 D is related to Fig. 5, B and D and Fig. S4, B and C. Fig. S4 E shows the STED microscopic images of co-stained HA-CERT and V5-C10orf76. Fig. S4 F shows the result of co-immunoprecipitation analysis of HA-C10orf76 and HA-CERT. The uncropped images of the immunoblots and STED microscopic observations are provided in Source Data files. Table S1 shows fold enrichment of sgRNAs. Table S2 shows raw data from the lipidomic analysis (sphingolipids). Table S3 shows raw data from the lipidomic analysis (glycerophospholipids). Lists of chemical reagents and antibodies can be found in Tables S4 and S5, respectively. The sequences of sgRNA targets, primers, and oligo DNAs are listed in Tables S6, S7, and S8, respectively. The plasmids used in this study are listed in Table S9.

### Data availability
The raw and uncropped data, which are not included in the supplemental material or source data, have been posted on a public data repository (https://doi.org/10.6084/m9.figshare.22315159).

## Acknowledgments
We would like to thank Yoshiyuki Sekizawa (Zenyaku Kogyo Co., Tokyo, Japan) for providing the lysenin, Minetaro Arita (Department of Virology II, National Institute of Infectious Diseases, Tokyo, Japan), and Norito Tamura (Department of Biochemistry and Cell Biology, National Institute of Infectious Diseases, Tokyo, Japan) for providing the plasmids. We also thank Chisato Sakuma (Department of Bacteriology I, National Institute of Infectious Diseases, Tokyo, Japan) for helping with the data acquisition for the next-generation sequencing and Kentaro Shimasaki (Department of Biochemistry and Cell Biology, National Institute of Infectious Diseases, Tokyo, Japan) for providing HeLa cells stably expressing HA-CERT.

This work was supported by Sasakawa Scientific Research Grant from the Japan Science Society (grant number 2022-4046) to A. Mizuike, Japan Agency for Medical Research and Development (AMED; JP20he0622012) and Ministry of Education, Culture, Sports, Science and Technology (MEXT) KAKENHI (JP17H06413 and JP17H06417) to K. Katoh, Japan Society for the Promotion of Science (JSPS) KAKENHI (JP17K07357 and JP21H02435) to T. Yamaji and MEXT KAKENHI (JP17H06417), AMED-CREST (P20gm0910005j0006), and JSPS KAKENHI (JP21H02630) to K. Hanada. Open Access funding provided by National Institute of Infectious Diseases.

Author contributions: Conceptualization, K. Hanada; Methodology, A. Mizuike, S. Sakai, K. Katoh, and T. Yamaji; Formal analysis, A. Mizuike and T. Yamaji; Investigation, A. Mizuike, S. Sakai, and K. Katoh; Writing, A. Mizuike. and K. Hanada; Funding Acquisition: A. Mzuike, K. Katoh, T. Yamaji, and K. Hanada. Supervision, K. Hanada.

Disclosures: The authors declare no competing interests exist.

Submitted: 17 November 2021

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

# Supplemental material

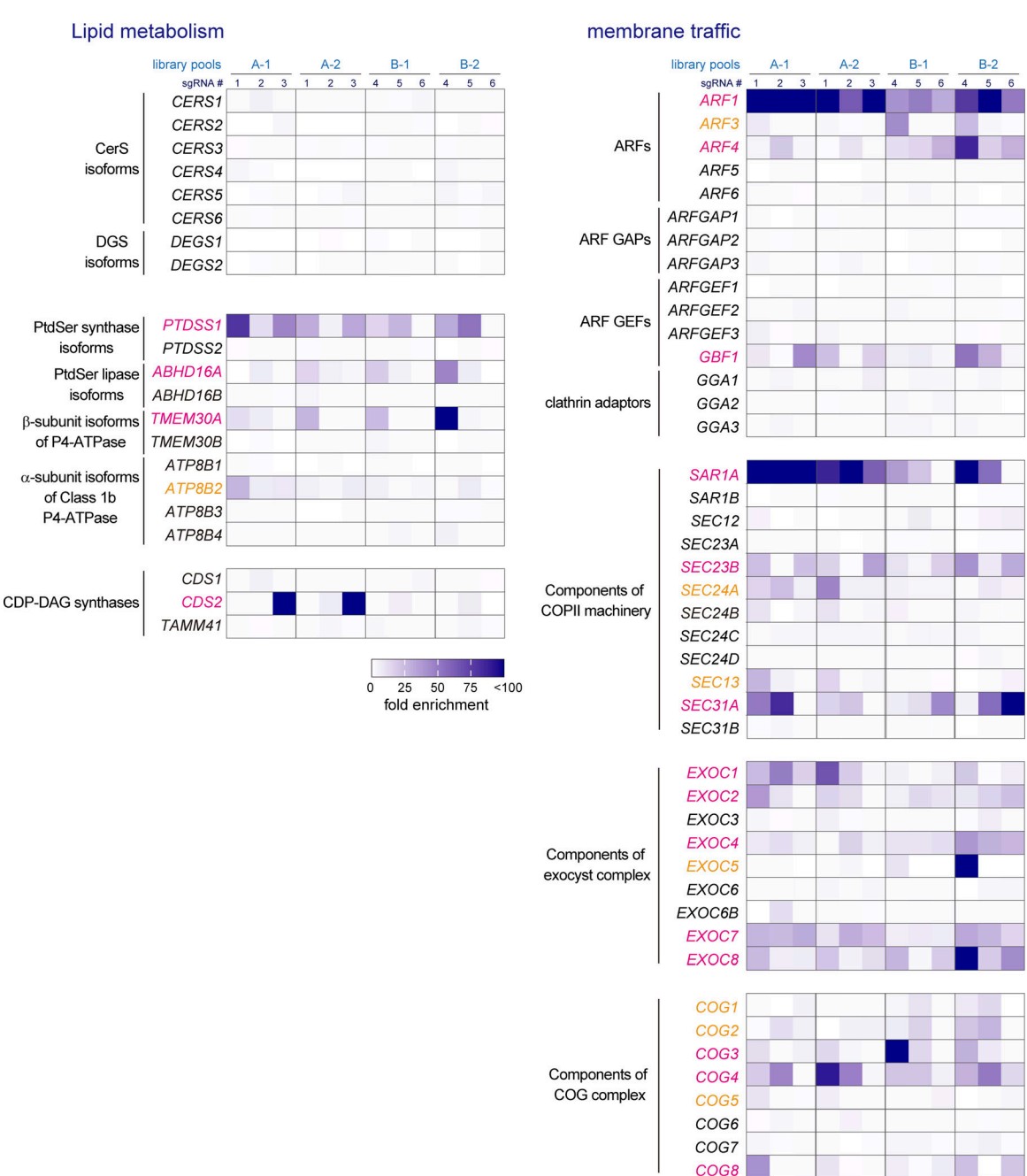

Figure S1. **Identification of genes conferring lysenin resistance by disruption.** Heatmaps of representative genes involved in lipid metabolism and membrane traffic are shown. Heatmaps depict the fold-enrichment of normalized read numbers of six sgRNAs of each gene in two independent experiments. Genes enriched during the screening are highlighted in color: magenta represents genes for which sgRNAs were enriched in both the A and B libraries, and orange represents those enriched in just one of the libraries.

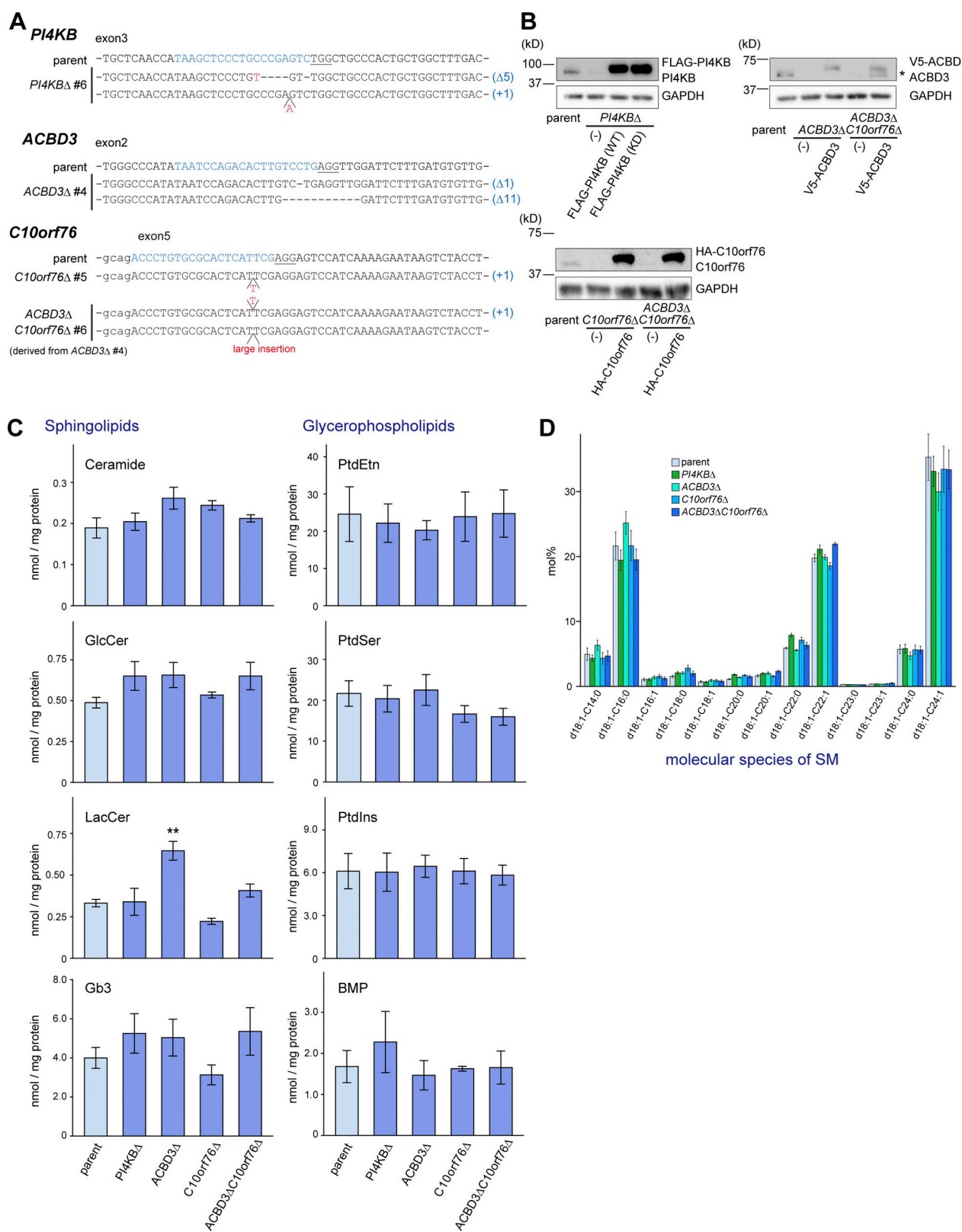

Figure S2. **Establishment of the KO cell clones and lipidomic analysis. (A)** Sequence analysis of genomic DNA derived from genome-edited clones. **(B)** Confirmation of KO by Western blot analysis. Cell lysates were immunoblotted with anti-PI4KB, anti-ACBD3, anti-C10orf76, or anti-GAPDH antibodies. *, degradation product. **(C and D)** Lipidomic analysis of the KO cells (supplemental information for Fig. 2 C). Lipids were extracted from cells subcultured in FBS-free medium and analyzed by LC–MS/MS. The bar graph represents the mean ± SEM of three independent experiments. Statistical significance was determined by two-sided Dunnett test. *, P < 0.05. **(D)** The molecular species of SM was unaltered by gene disruption. Source data are available for this figure: SourceData FS2.

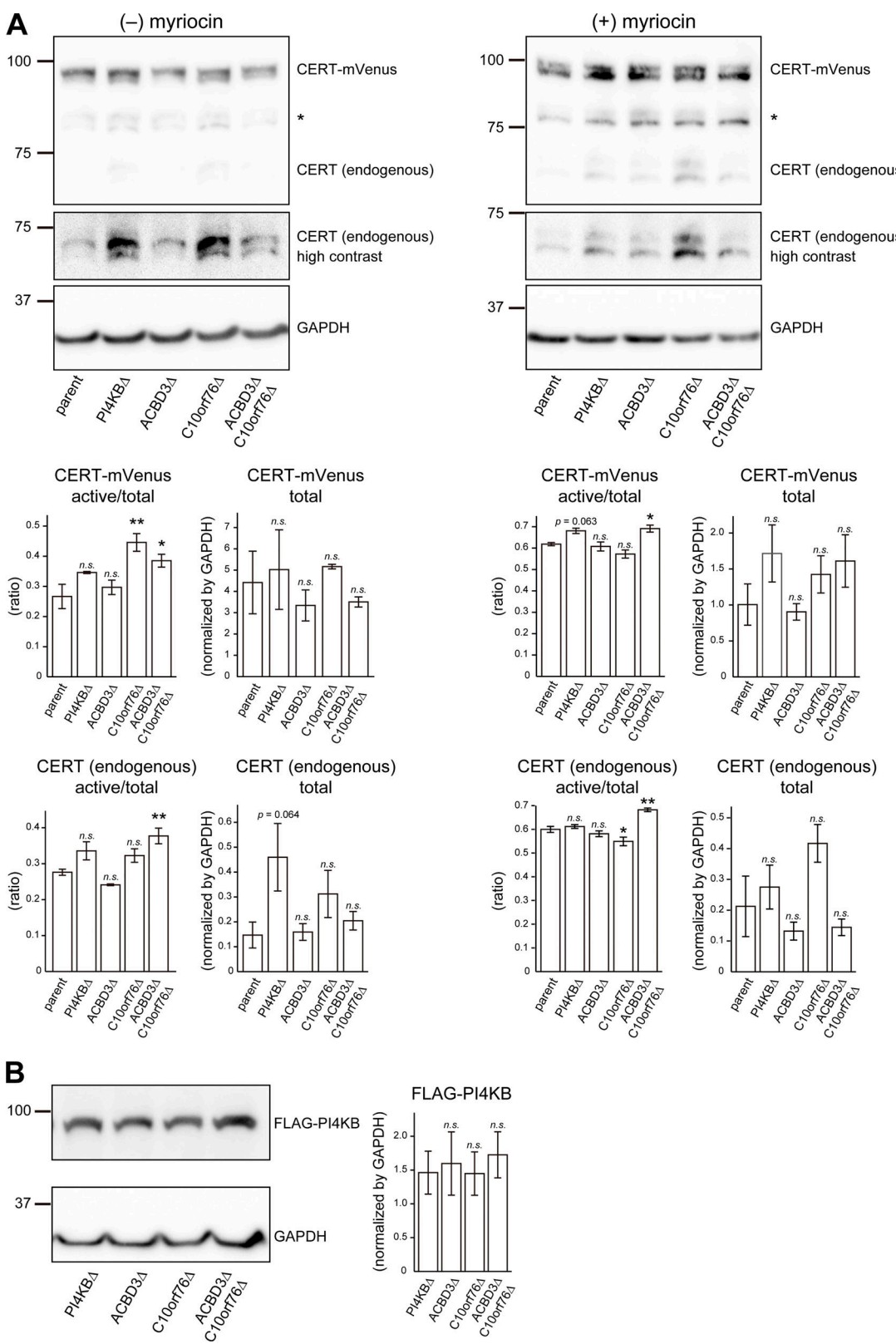

**Figure S3.** **The expression levels of recombinant proteins expressed in the KO cells are similar to those in the parent cells. (A)** Western blot analysis of the endogenous CERT and ectopic CERT-mVenus expressed in various cells treated with or without the sphingolipid synthesis inhibitor myriocin. Cells stably expressing CERT-mVenus were cultured with or without 2.5 µM myriocin for 24 h. Cell lysates were immunoblotted with anti-CERT or anti-GAPDH antibodies. Statistical significance was determined by two-sided Dunnett test. *, P < 0.05, **, P < 0.01. **(B)** Western blot analysis of FLAG-PI4KB stably expressed in various cells. Cell lysates were immunoblotted with anti-FLAG or anti-GAPDH antibodies. Statistical significance was determined by two-sided Dunnett test. n.s., not significant. Source data are available for this figure: SourceData FS3.

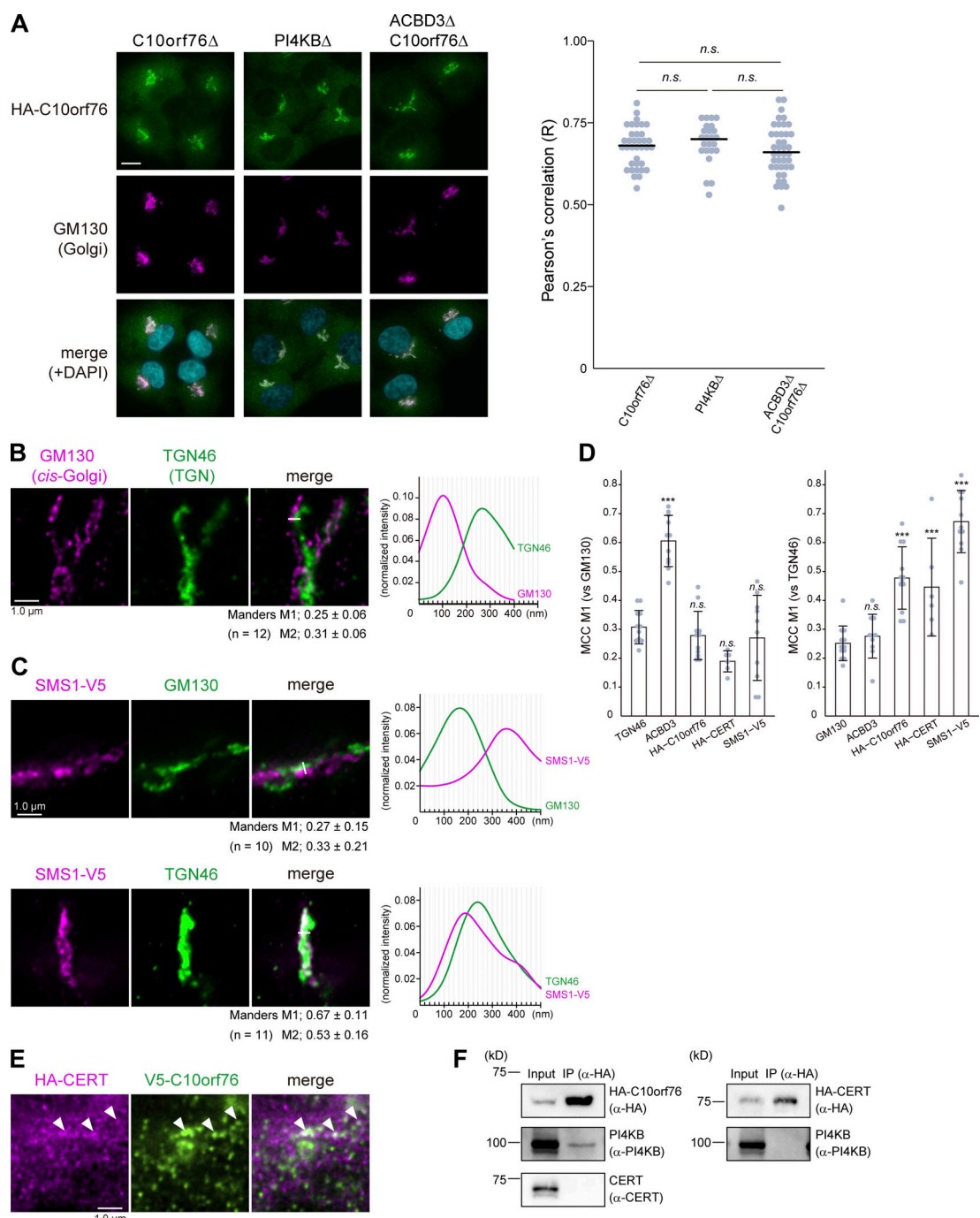

Figure S4.   **Only a portion of CERT colocalizes with C10orf76. (A)** Microscopic observation of the intracellular distribution of HA-C10orf76 in KO cells. Cells stably expressing HA-C10orf76 were fixed and immunostained with anti-HA and anti-GM130 antibodies. Nuclei were visualized with DAPI. Scale bar, 10 µm. The Pearson's correlation coefficients between HA-C10orf76 and the Golgi marker GM130 in one cell (*n* = 24–41), calculated from at least three images, are shown on the right. The line segments represent the median. Statistical significance was determined by Steel-Dwass test. n.s., not significant. Representative data from at least two independent experiments with similar results are shown. **(B)** STED microscopic observation of cis-Golgi (marked by GM130) and the TGN (marked by TGN46). Fixed cells were immunostained with anti-GM130 and anti-TGN46 antibodies. **(C)** STED microscopic observation of the intra-Golgi distribution patterns of SMS1-V5. Cells stably expressing SMS1-V5 were fixed and immunostained with anti-V5 + anti-GM130 antibodies or anti-V5 + anti-TGN46 antibodies. Graphs represent the line profiles of regions of interest (ROIs, depicted by white lines): x-axis, distance; y-axis, normalized signal intensity. **(B and C)** The mean ± SD of Manders correlation coefficient (MCC) and the cell numbers (*n*) analyzed are shown. MCC M1 represents the colocalization rate of the protein of interest with the marker protein. M2 represents the colocalization rate of the marker protein with the protein of interest. **(D)** The mean ± SD of MCC M1 from Fig. 5, B and D, and Fig. S4, B and C are shown. Dots represent the individual data points. Statistical differences between the marker protein (TGN46 or GM130) and the proteins of interest are shown. Statistical significance was determined by one-sided Dunnett test. ***, P < 0.001; n.s., not significant. **(E)** STED microscopic observation of HA-CERT and V5-C10orf76. **(B, C, and E)** Scale bars, 1.0 µm. **(F)** Co-immunoprecipitation analysis of CERT, C10orf76, and PI4KB. Cell lysates, prepared from cells stably expressing HA-C10orf76 or HA-CERT, were subjected to immunoprecipitation with anti-HA-agarose. The eluates were immunoblotted with anti-HA, anti-PI4KB, or anti-CERT antibodies. Source data are available for this figure: SourceData FS4.

Provided online are Data S1, Table S1, Table S2, Table S3, Table S4, Table S5, Table S6, Table S7, Table S8, Table S9. Data S1 displays CRISPR_process.pl. Table S1 shows fold enrichment of sgRNAs. Table S2 shows raw data from the lipidomic analysis (sphingolipids). Table S3 shows raw data from the lipidomic analysis (glycerophospholipids). Table S4 lists chemical reagents. Table S5 lists antibodies. Table S6 shows the DNA sequences encoding sgRNAs. Table S7 lists primers used for CDS amplification or base substitution. Table S8 lists the sequences of oligo DNA. Table S9 lists the plasmids used in this study.

