## [Peer Review File · The Journal of Cell Biology]

The C10orf76-PI4KB axis orchestrates CERT-mediated ceramide trafficking to the distal Golgi

Aya Mizuike, Shota Sakai, Kaoru Katoh, Toshiyuki Yamaji, and Kentaro Hanada

Corresponding Author(s): Kentaro Hanada, National Institute of Infectious Diseases

Review Timeline:

Submission Date:	2021-11-17
Editorial Decision:	2021-11-19
Revision Received:	2022-06-01
Editorial Decision:	2022-08-15
Revision Received:	2022-12-16
Editorial Decision:	2023-02-03
Revision Received:	2023-03-04
Editorial Decision:	2023-03-17
Revision Received:	2023-03-23

Monitoring Editor: Tamas Balla

Scientific Editor: Tim Spencer

Transaction Report:

DOI: <https://doi.org/10.1083/jcb.202111069>

November 19, 2021

Re: JCB manuscript #202111069

Dr. Kentaro Hanada
National Institute of Infectious Diseases
Department of Biochemistry & Cell Biology
1-23-1 Toyama, Shinjuku-ku
Tokyo 1628640

Dear Dr. Hanada,

Thank you for submitting your Article manuscript entitled "ACBD3 and C10orf76 determine the distinct Golgi contact zones for inter-organelle ceramide transport" to Journal of Cell Biology. As part of our normal reviewing procedure, your paper has been evaluated by at least two editors and an editorial statement is provided below. You will see that, in the consensus opinion of our editors, the manuscript is not a good fit for Journal of Cell Biology. We have thus decided not to subject your manuscript to a lengthy external review process. Because Journal of Cell Biology addresses a wide and diverse audience of cell biologists, we must give priority to manuscripts that provide a substantial advance of broad appeal to the cell biology community, even though many others also present interesting and important advances for researchers in a particular field.

I am sorry that our answer on this occasion is not more positive, and I hope that this outcome will not dissuade you from submitting other manuscripts to us in the future.

Thank you for your interest in Journal of Cell Biology.

With kind regards,

Jodi Nunnari
Editor-in-Chief
Journal of Cell Biology

Editorial Statement:

In this study, the authors describe CERT and ceramide transport as a central hub where the known PI4KB interacting proteins ACBD3 and C10orf76 intersect. This provides potentially interesting functional insight into the relatively poorly understood biological relationship between these proteins. However, while an interesting start the study currently does not provide sufficient information into the molecular relationship of these components as they support CERT function. One major issue pertains to the lack of a quantitative assessment of the localization studies, in particular to support to what extent ACBD3 and C10orf76 determine PI4KB Golgi localization and how they affect PI4P levels. The authors should also examine PI4K2A localization in knockout cells to rule out if its relocalization may be responsible for the unchanged OSBP localization. Another consideration is why PIK93 or myriocin treatment changes the localization of PI4KB and how that relates to the interaction among the three proteins. While the description of the disparate localization of ACBD3 and C10orf76 by STED are appreciated, it is unclear how that explains the additive effects on CERT function and support that CERT is more closely aligned with C10orf76 than ACBD3 is lacking. Finally, the discrepancy with the McPhail study suggesting that C10orf7 is recruited by PI4KB, thus may act downstream, also needs to be addressed. Therefore, the study is currently not competitive for review in JCB.

August 15, 2022

Re: JCB manuscript #202111069R-A

Dr. Kentaro Hanada
National Institute of Infectious Diseases
Department of Quality Assurance, Radiation Safety & Information System
1-23-1, Toyama, Shinjuku-ku
Tokyo 162-8640
Japan

Dear Dr. Hanada,

Thank you for submitting your manuscript entitled "The C10orf76-PI4KB axis orchestrates CERT-mediated ceramide trafficking to the distal Golgi". The manuscript was assessed by expert reviewers, whose comments are appended to this letter. We invite you to submit a revision if you can address the reviewers' key concerns, as outlined here.

As you will see, the reviewers find your results regarding functionally distinct pools of PI4P of potential high interest for the readership of JCB. However, they have indicated several important points where the data need to be strengthened to better support your model. In particular, we agree with reviewers 1 and 3 that better localization data is required and that this should therefore be a major focus in revising. Notably, while your model proposed localization as a means of regulation, at this point it does not seem alternative contributions such as protein-protein associations, conformational changes, or post-translational modifications such as phosphorylation, can be excluded. Therefore, it is essential to improve the localization data and analysis of Golgi morphology, and if these suggest the contribution of additional or alternative factors this must be included in your interpretation and discussion.

We appreciate that there are no antibodies to stain endogenous CERT, therefore this is not required. If it is possible to test the phosphorylation status of CERT, for example via IP or mass-spec, we think this would greatly increase the value of your study. However, it is not essential if this is not technically feasible. You must also address the question of reviewer 2 regarding how to distinguish sphingomyelin synthesis defects and rule out post-Golgi SM trafficking problems, though this can be done at the level of writing. In addition, we hope that you will be able to address all of the remaining reviewer comments in your revised manuscript.

GENERAL GUIDELINES:

Text limits: Character count for an Article is < 40,000, not including spaces. Count includes title page, abstract, introduction, results, discussion, and acknowledgments. Count does not include materials and methods, figure legends, references, tables, or supplemental legends.

Figures: Articles may have up to 10 main text figures. Figures must be prepared according to the policies outlined in our Instructions to Authors, under Data Presentation, <https://jcb.rupress.org/site/misc/ifora.xhtml>. All figures in accepted manuscripts will be screened prior to publication.

Supplemental information: There are strict limits on the allowable amount of supplemental data. Articles may have up to 5 supplemental figures. Up to 10 supplemental videos or flash animations are allowed. A summary of all supplemental material should appear at the end of the Materials and methods section.

Please note that JCB now requires authors to submit Source Data used to generate figures containing gels and Western blots with all revised manuscripts. This Source Data consists of fully uncropped and unprocessed images for each gel/blot displayed in the main and supplemental figures. Since your paper includes cropped gel and/or blot images, please be sure to provide one Source Data file for each figure that contains gels and/or blots along with your revised manuscript files. File names for Source Data figures should be alphanumeric without any spaces or special characters (i.e., SourceDataF#, where F# refers to the associated main figure number or SourceDataFS# for those associated with Supplementary figures). The lanes of the gels/blots should be labeled as they are in the associated figure, the place where cropping was applied should be marked (with a box),

and molecular weight/size standards should be labeled wherever possible.

The typical timeframe for revisions is three to four months. While most universities and institutes have reopened labs and allowed researchers to begin working at nearly pre-pandemic levels, we at JCB realize that the lingering effects of the COVID-19 pandemic may still be impacting some aspects of your work, including the acquisition of equipment and reagents. Therefore, if you anticipate any difficulties in meeting this aforementioned revision time limit, please contact us and we can work with you to find an appropriate time frame for resubmission. Please note that papers are generally considered through only one revision cycle, so any revised manuscript will likely be either accepted or rejected.

Thank you for this interesting contribution to Journal of Cell Biology. You can contact us at the journal office with any questions, cellbio@rockefeller.edu or call (212) 327-8588.

Sincerely,

Tamas Balla, MD, PhD
Monitoring Editor

Andrea L. Marat, PhD
Senior Scientific Editor

Journal of Cell Biology

Reviewer #1 (Comments to the Authors (Required)):

This study uses a clever CRISPR/Cas9 KO screen to identify KOs that reduce SM levels in cells. It finds that SM levels are reduced in KOs of the PI 4-kinase PI4KB and the PI4KB-binding proteins ACBD3 and C10orf76. Evidence is provided that PI4KB is necessary to support CERT-facilitated SM synthesis by enriching CERT at contacts between the ER and Golgi. The study goes on to argue that C10orf76 brings CERT to distal Golgi regions, which are thought to be the site of SM synthesis. C10orf76-mediated enrichment of PI4KB at the distal Golgi could thus support metabolic channeling of ceramide to SM synthase by CERT. These findings could explain previous work that suggests CERT supports the synthesis of SM but not GlcCer. Overall, the study is well done, but more work is necessary to support the model.

1. More evidence is necessary to support the claim that PI4KB, ACBD3, and C10orf76 KO alters the amount of CERT on the Golgi. The results in Fig 3A suggest CERT is more abundant in the KO cell lines. If that is correct, then the modest changes in Golgi enrichment shown in Fig 3C might not indicate a significant change in the amount of CERT at the Golgi. Perhaps only the size of the cytoplasmic pool changes.

2. It would be good if the results in Fig 4D were verified with a second PI4P sensor. More importantly, if the model is correct PI4P should be specifically enriched on the parts of the Golgi containing SMS1. This should be determined. If PI4P is enriched in these regions, it is important to determine whether the degree of enrichment is sufficient to significantly increase CERT binding. It seems unlikely that the modest change in PI4P levels in C10orf76 KO cells would significantly affect CERT binding.

3. The Pearson colocalization coefficient for the results in Fig. 5 should be shown and discussed.

4. It is not clear what the experiments that show localization of VAP-A with CERT and other proteins add to the study (Fig. 6C). In fact, the results somewhat undercut the model. If VAP-A is necessary to bring CERT to the ER, shouldn't there be a significant overlap of VAP-A and CERT signal? Since there is not, perhaps most of the CERT is not at ER contact sites.

Reviewer #2 (Comments to the Authors (Required)):

Review of C10orf76-PI4KB axis for ceramide transport

In this manuscript the authors proposed a mechanism whereby CERT can be directed to specific Golgi areas where PI4P is synthesized by PI4KB associated with C10orf76. To find this they performed a lysenin resistance screen, similar in some respects to how they found CERT in the past, but now using CRISPR. The screen is a rather elegant way to find these "accessory" factors which, in addition to CERT, are required to transport ceramide to the correct place. It is important to understand how CERT activity is targeted to the Golgi because we know that CERT transferred ceramide is used to synthesize SM, but not GlcCer. Another aspect of this work is that we know that PI4P is found elsewhere in the Golgi, but here they come up with an explanation of how CERT can be directed to the correct location. It is only recognizing the PI4P synthesized with the help of C10orf67. As PI4P is probably indistinguishable if synthesized using C10orf67 or ACBD3 (we do not have any data on acyl chain composition so we cannot be 100% sure), this implies a protein-protein interaction between CERT and C10orf67. They show a colocalization and I assume that proximity labeling techniques would show a close association, but there is no evidence for a direct protein-protein interaction. A future experiment could be to look for such an interaction, either by co-immunoprecipitation or in vitro reconstitution. There is no guarantee that this type of experiment will work. So while this work gives good evidence for distinct functions of PI4P in different regions of the Golgi and implicates C10orf67 in this mechanism, the question of whether it is a distinct pool of PI4P or a coincidence detection of PI4P with an accessory protein is not resolved. This could be stated more clearly and discussed more thoroughly.

Some points on the figures where I was not totally convinced or had questions about the data.

Figure 2: Why does the ACBD3/c10orf76 double mutant restore viability completely under 100 ng/ml lysenin, but not with 50 ng/ml? Also the double is as efficient as the PI4KB KO at 100, but less at 50?

While the data of SM amounts do correlate well with viability it might be worth a comment about why such relatively small decreases give such big differences in viability. Does this correlate with some mechanistic aspect of lysenin killing? I assume that once SM is made after CERT delivery of ceramide that the SM can be delivered normally to the cell surface.

Figure 3. Western blotting analysis showed that gene disruption of PI4KB, ACBD3, or C10orf76 resulted in no discernible impact on the doublet pattern of endogenous CERT, with the hyperphosphorylated form being dominant (Fig. 3 A). This statement does not seem to correspond to the data in figure 3A. It looks to me that the unphosphorylated CERT band is more pronounced in the C10orf67 KO.

Reviewer #3 (Comments to the Authors (Required)):

In this study, Mizuike et al. investigate the molecular mechanisms involved in ceramide transport from the ER to the Golgi by the ceramide transport protein CERT. They used a genome-wide CRISPR/Cas9 KO screen to identify genes for which KO confers Lysenin resistance. Lysenin binds to sphingomyelin at the plasma membrane and creates pores that lead to cell death.

Trafficking of ceramide from the endoplasmic reticulum (ER) to the Golgi is required for sphingomyelin synthesis and, therefore, its trafficking to the plasma membrane. With this screen, the authors identified a number of genes for which KO confers lysenin resistance and that are thus potentially involved in ceramide trafficking. Among these genes, the authors focused on genes associated with PI4P regulation, which included PI4KB, a PI4P kinase that produces PI(4)P at the Golgi, and ACBD3 and C10orf76, which encode proteins interacting with PI4KB. The authors then tested whether these genes were required for the recruitment of the ceramide transfer protein CERT at ER-Golgi contact sites. They first confirmed that PI4KB, ACBD3 and C10orf76 depletion confers lysenin resistance and that they are important for sphingomyelin synthesis. They then investigated whether these proteins were required for PI4KP and CERT localization and function to the Golgi. The authors then used super-resolution imaging to look for the localization of ACBD3 and C10orf76 at different Golgi compartments and observed that ACBD3 localizes more to proximal Golgi (Ci-Golgi) and that C10orf76 localizes more to distal Golgi (Trans-Golgi) and more precisely ER-Trans-Golgi contact sites where ceramide transfer and sphingomyelin synthesis occur.

This manuscript is clearly written, and the experiments were well described in text/figures. Also, the quality of the experiments was high quality. Moreover, this study shows an important role for the C10orf76-PI4KB axis that notably produces PI(4)P at the Golgi in sphingomyelin trafficking to the plasma membrane, which requires ER to Golgi ceramide transfer. This would be of interest to the readers of JCB.

However, I have some major concerns with some of the interpretations of their data made by the authors as alternative interpretations were not addressed, and for others, control experiments are missing. Together, these issues make it difficult to conclude that ACBD3 and C10orf76 are required for Golgi recruitment of PI4KB for CERT activation.

Major points:

1) The authors conclude that PI4KB, ACBD3 and C10orf76 are important for CERT localization at the Golgi and that ACBD3 and C10orf76 are important for PI4KB recruitment at the Golgi. However, the Golgi enrichment of CERT-mVenus (fig 3B-c) and Flag-PI4KB (fig 4B-c) does not show any impairment at basal conditions and only when their localization is being pharmacologically forced (Myriocin for CERT and PIK93 for PI4KB). One would expect to see a defect at basal condition if any of these proteins are required for CERT or PI4KB recruitment to the Golgi Apparatus. There are several possible reasons for this. i) First, looking closely at the MG130 staining, the Golgi Apparatus looks different between the various KO, with ACBD3-del appearing more condensed. In fact, the GM130 staining in Figure 4B appears to be missing in the PI4KB-del and C10orf76-del cells. Since GM130 was used as the template for measuring the Golgi apparatus area, the difference in the Golgi may affect the enrichment

quantification. ii) In Figure 3A, the authors show that CERT expression was increased in the KO cell lines. Could this increase in endogenous CERT be completed with the ectopic expression of CERT-mVenus? iii) CERT and PI4KB enrichment measurements are the ratio of the CERT-mVENUS or FLAG-PI4KB signal intensity in Golgi localized vs the cytosol. For this quantification, the assumption is that the ectopically expressed cells between the different cell lines have the same expression. Has this been quantified? To resolve some of these issues, the authors should use antibodies against endogenous proteins. I am not sure whether the CERT antibody use in 3A is feasible for IF, but the PI4KB should be feasible for these studies.

2) For the CERT and PI4KB enrichment quantification, is GM130 a good marker given that PI4KB may be localized to the trans-Golgi? Further, another protein that localization at the Golgi should be used as a negative control to demonstrate and validate their quantification. Similarly, using Brefeldin A treatment which was shown to abolish PI4KB localization at the Golgi, should also be used as a control.

3) Based on their western blot of CERT in Figure 3A, the author stated that the knockout had "...no discernible impact on the doublet pattern of endogenous CERT, with the hyperphosphorylation from being dominant." Although I would agree that there is a higher ratio of the hyperphosphorylation band compared to the lower band for most, this is not the case for C10orf76A-del, where the lower band appears higher. Wouldn't this be expected if C10orf76A is required for SM production, as seen in figure 2A? In fact, wouldn't you see less hyperphosphorylated band in all the KO cell lines, given that SM is lower?

4) One important control lacking for all the KO cell lines is a morphologic analysis of the Golgi to show that the overall Golgi morphology is not affected by the various knockouts or treatments in a way that would affect the results of colocalization measures.

5) The authors provide very robust data indicating that deletion of PI4KB, C10orf76 or ACBD3 impairs sphingomyelin synthesis and levels at the plasma membrane but do not show controls that the lysenin resistance conferred by these deletions is primarily due to a defect in ceramide transfer and not sphingomyelin trafficking. Such controls should be performed. For instance, confirming the ER-to Golgi ceramide trafficking defect by using fluorescent ceramide or ceramide antibodies would strengthen the conclusions. Similarly, validating sphingomyelin synthesis defects by using a sphingomyelin biosensor (equinatoxin) or antibodies would support the claims of the authors.

Minor points:

- Fig1B: If possible, a complementation assay performed with PI4KB mutants unable to interact with either ACBD3 or C10orf76 could be an interesting experiment to perform in order to support the different roles of these two proteins in PI4KB recruitment/function at the Golgi.
- Fig3A: the authors should provide a quantification of the immunoblot comparing levels of CERT but also the ratio phosphorylated/non-phosphorylated of CERT in the different conditions. Especially since the state in the text that they did not observe any discernible effect on the doublet (with the upper band being dominant) pattern between conditions. The image shown here seems to indicate that the ratio between phosphor and non-phospho band changes between conditions and that maybe the lower band appears dominant in C10orf76 KO cells. The authors should quantify that and (if there is a change) comment/discuss it in the text.
- Fig3C and other Golgi enrichment measures: It is not clear in the text, figure or legends what is the units of Golgi enrichment that are measured/shown here (arbitrary units (but how they were defined?), percentage of colocalization? etc.). This should be indicated in the figures and in the legends.
- Fig 3C: The authors should make it obvious that the statistical analysis was performed on Myriocin treated cells.
- Fig 3D: insets are particularly small; it would help the reader if their size is increased.
- Fig4A: the blot should be quantified as well.
- Fig4D: The fact that only the P values below 0.05 are shown that all other pairs are $p < 0.05$ does not appear clear on the graph. This should be modified to make it more obvious what is considered statistically significant or not.
- Fig5C is particularly dim compared to other panels. Brightness should be increased to facilitate visualization by readers.
- Fig6a: the legend refers to Figure S5 for the uncropped image, but only 3 supplemental figures were provided.
- Line 312-314: the authors refer to Fig4C, while it is Fig4D that should be referred to.
- Line 461: It appears that one word is missing in this sentence. In its current version, this sentence does not mean anything.

Dear Dr. Balla, Editor of the Journal of Cell Biology,

We appreciate the time and effort that you and the reviewers have dedicated to providing feedback on our manuscript and are grateful for the insightful comments. We have incorporated most of the suggestions from the editor and reviewers. These changes have been highlighted within the manuscript. Below, please find our responses to the reviewers' comments, addressed in a point-by-point manner.

I hope that our revised manuscript appropriately addresses the issues raised by the reviewers.

Sincerely,

Reviewer #1 (Comments to the Authors):

This study uses a clever CRISPR/Cas9 KO screen to identify KOs that reduce SM levels in cells. It finds that SM levels are reduced in KOs of the PI 4-kinase PI4KB and the PI4KB-binding proteins ACBD3 and C10orf76. Evidence is provided that PI4KB is necessary to support CERT-facilitated SM synthesis by enriching CERT at contacts between the ER and Golgi. The study goes on to argue that C10orf76 brings CERT to distal Golgi regions, which are thought be the site of SM synthesis. C10orf76-mediated enrichment of PI4KB at the distal Golgi could thus support metabolic channeling of ceramide to SM synthase by CERT. These findings could explain previous work that suggests CERT supports the synthesis of SM but not GlcCer. Overall, the study is well done, but more work is necessary to support the model.

1. More evidence is necessary to support the claim that PI4KB, ACBD3, and C10orf76 KO alters the amount of CERT on the Golgi. The results in Fig 3A suggest CERT is more abundant in the KO cell lines. If that is correct, then the modest changes in Golgi enrichment shown in Fig 3C might not indicate a significant change in the amount of CERT at the Golgi. Perhaps only the size of the cytoplasmic pool changes.

Author response: Thank you very much for your critical and constructive comments. To address your concerns, we added the image analysis of immunoblots of the endogenous CERT and ectopically expressed CERT-mVenus (Fig. 3 A and Fig. S3 A). Interestingly, the protein level of the endogenous CERT, but not the ectopic CERT-mVenus, was

up-regulated when *PI4KB* was disrupted, suggesting the presence of a transcriptional regulatory system of *CERT1* in response to the dysfunction of CERT. It should also be pointed out that the expression level of CERT-mVenus was similar in all host cell lines (Fig. 3 B and C). This eliminated the possibility that the modest changes in Golgi enrichment of CERT-mVenus could be accounted for only by changes in its expression level. We added the following sentences as an explanation (p. 18, l. 300): “Note that the expression levels of CERT-mVenus in the KO cells were comparable to that of the parent cells (Fig. S3 A). Nonetheless, the KO of *C10orf76* still increased the ratio of the de- or hypo-phosphorylated form of CERT-mVenus, and the expression levels and the phosphorylation state of endogenous CERT in the KO cells tended to be unaltered by ectopically expressing CERT-mVenus (Fig. S3 A, see the contrasted image).”

2. It would be good if the results in Fig 4D were verified with a second PI4P sensor. More importantly, if the model is correct PI4P should be specifically enriched on the parts of the Golgi containing SMS1. This should be determined. If PI4P is enriched in these regions, it is important to determine whether the degree of enrichment is sufficient to significantly increase CERT binding. It seems unlikely that the modest change in PI4P levels in C10orf76 KO cells would significantly affect CERT binding.

Author response: We attempted to employ P4M-SidM as another PtdIns(4)P probe (Hammond et al., *J Cell Biol*: 2014, **205**, 113-126). However, we found that its expression caused substantial morphological changes in the Golgi apparatus and the formation of vacuoles in HeLa cells (see the data for the reviewer only). For this reason, we could not use P4M-SidM for quantitative monitoring of the Golgi PtdIns(4)P level in our study. It would be of interest to observe PtdIns(4)P specifically enriched on the SMS1- (or CERT-) positive Golgi regions. However, there remain technical problems in observing PtdIns(4)P by super-resolution microscopy. It has been reported that the lipid molecules cannot be fully fixed by the present fixation method, and a modest movement of lipid molecules in the fixed samples is inevitable (Tanaka et al., *Nat. Methods*: 2010, **7**, 865–866). Optimization of the protocols to prepare specimens to observe PtdIns(4)P with super-resolution microscopy will be a future task.

3. The Pearson colocalization coefficient for the results in Fig. 5 should be shown and discussed.

Author response: We added the Pearson's correlation coefficient for Fig. 5 A and Manders colocalization coefficient for Fig. 5 B, D and Fig. S4 C and discussed the intra-Golgi distribution of the proteins of interest. The STED images of Fig. 6 A, C and Fig. S4 D were inadequate to compare the Mander's coefficients due to the high cytosolic signals. Thus, we revised the manuscript as follows (p. 23, l. 383): "Manders colocalization coefficient M1, calculated from the uncropped images, verified that ACBD3 colocalized more with GM130 than TGN46, and C10orf76 colocalized more with TGN46 than GM130."

4. It is not clear what the experiments that show localization of VAP-A with CERT and other proteins add to the study (Fig. 6C). In fact, the results somewhat undercut the model. If VAP-A is necessary to bring CERT to the ER, shouldn't there a significant overlap of VAP-A and CERT signal? Since there is not, perhaps most of the CERT is not at ER contact sites.

Author response: In the perinuclear regions, HA-CERT signals frequently, even if not perfectly, overlapped with or located in the proximity of VAP-A signals (overlapping signals are indicated by arrowheads). In contrast, SMS1-V5 displayed a broader distribution than HA-CERT in the Golgi complex and was not enriched in the vicinity of VAP-A (Fig. 6 A and C), ruling out the possibility that proteins embedded in the distal Golgi were non-specifically enriched in the vicinity of VAP-A. These results suggest that the majority of Golgi-distributed CERT molecules were binding to VAP-A at the ER-distal Golgi contact sites, in line with our previous study (Kawano et al., *JBC*: 2006). It should also be pointed out that we used the N-terminal HA-tagged CERT in the STED microscopy experiments and that the HA-tag site was adjacent to the Golgi-binding PH domain, which was ~330 amino acids far from the VAP-binding FFAT-motif in CERT. This might have caused a non-perfect overlap of the distal Golgi-distributed HA-CERT with VAP-A at the super-resolution level.

Reviewer #2 (Comments to the Authors):

In this manuscript the authors proposed a mechanism whereby CERT can be directed to specific Golgi areas where PI4P is synthesized by PI4KB associated with C10orf76. To find this they performed a lysenin resistance screen, similar in some respects to how they found CERT in the past, but now using CRISPR. The screen is a rather elegant way to find these "accessory" factors which, in addition to CERT, are required to transport ceramide to the correct place. It is important to understand how CERT activity is

targeted to the Golgi because we know that CERT transferred ceramide is used to synthesize SM, but not GlcCer. Another aspect of this work is that we know that PI4P is found elsewhere in the Golgi, but here they come up with an explanation of how CERT can be directed to the correct location. It is only recognizing the PI4P synthesized with the help of C10orf67. As PI4P is probably indistinguishable if synthesized using C10orf67 or ACBD3 (we do not have any data on acyl chain composition so we cannot be 100% sure), this implies a protein-protein interaction between CERT and C10orf67. They show a colocalization and I assume that proximity labeling techniques would show a close association, but there is no evidence for a direct protein-protein interaction. A future experiment could be to look for such an interaction, either by co-immunoprecipitation or in vitro reconstitution. There is no guarantee that this type of experiment will work. So while this work gives good evidence for distinct functions of PI4P in different regions of the Golgi and implicates C10orf67 in this mechanism, the question of whether it is a distinct pool of PI4P or a coincidence detection of PI4P with an accessory protein is not resolved. This could be stated more clearly and discussed more thoroughly.

Some points on the figures where I was not totally convinced or had questions about the data.

Author response: Thank you very much for your critical and constructive comments. As per the comment “the question of whether it is a distinct pool of PtdIns(4)P or a coincidence detection of PtdIns(4)P with an accessory protein is not resolved”, we added the following sentences at p. 24, l. 402: “Of note, only a portion of HA-CERT and V5-C10orf76 significantly overlapped when observed by STED microscopy (Fig. S4 D). Moreover, there was no detectable co-immunoprecipitation of C10orf76 or PI4KB along with CERT (Fig. S4 D), while PI4KB was observed to precipitate along with C10orf76 (Fig. S4 E), which was consistent with previous studies (Greninger et al., 2013; McPhail et al., 2020). Thus, it is unlikely that CERT directly interacts with C10orf76 and/or PI4KB to be recruited to the distal Golgi regions.” We also discussed that “Of note, the co-immunoprecipitation assay suggested that C10orf76 or PI4KB did not directly interact with CERT (Fig. S4 E). In addition, no physical interactions between CERT (also known as COL4A3BP) and PI4KB or C10orf76 were found in a human protein–protein interaction database (<http://www.interactome-atlas.org/> assessed on 17 August 2022). STED microscopy observation also revealed that only a portion of CERT colocalized with C10orf76 (Fig. S4 D). Nonetheless, there remains the possibility that another accessory protein(s) is required for CERT to be recruited to the distal Golgi in a

C10orf76–PI4KB axis-dependent manner.” at p. 28, l. 473. We also addressed that “Proteins segregated or recruited to the PtdIns(4)P-rich microdomains are also likely to affect the local environment.” at p.31, l. 523.

We have responded to other specific comments as described below.

Figure 2: Why does the ACBD3/c10orf76 double mutant restore viability completely under 100 ng/ml lysenin, but not with 50 ng/ml? Also the double is as efficient as the PI4KB KO at 100, but less at 50?

Author response: We performed the experiment with 50 ng/mL and 100 ng/mL on different days, so the difference in the conditions may have affected the viability. To avoid confusion, we have replaced the data with newly analyzed data (Fig. 2 B).

While the data of SM amounts do correlate well with viability it might be worth a comment about why such relatively small decreases give such big differences in viability. Does this correlate with some mechanistic aspect of lysenin killing? I assume that once SM is made after CERT delivery of ceramide that the SM can be delivered normally to the cell surface.

Author response: To address your comment, we added a brief explanation at p. 13, l. 209 as follows: “Lysenin binds to clustered SM, not mono-dispersed SM molecules, in membranes (reviewed in Yilmaz et al., 2018), and only a partial decrease in cellular SM level is enough to gain discernible resistance to lysenin in mammalian cells (Hanada et al., 1998).”

The discrepancy commented on by the reviewer may be alternatively accounted for by the possibility that the dysfunction of PI4KB compromises the transport of SM from the distal Golgi to the PM. Thus, we also discussed this point as follows (p. 26, l. 435): “Metabolic labeling of lipids with radioactive serine and intracellular movement of C₅-DMB-ceramide provided compelling evidence that disrupting the C10orf76–PI4KB axis-compromises both *de novo* synthesis of SM and ER-to-Golgi transport of ceramide in living cells (Fig. 2 D and E). Acquiring the lysenin-resistance by disrupting the C10orf76–PI4KB axis via compromise of CERT function was in line with previous studies showing that the dysfunction of CERT endows lysenin-resistance to cells (Hanada et al. 2003; Yamaji & Hanada, 2014; Murakami et al., 2021). However, we do not deny the possibility that disrupting the C10orf76–PI4KB axis might also have

compromised the transport of SM from the distal Golgi to the PM in the present study.”

Figure 3. Western blotting analysis showed that gene disruption of PI4KB, ACBD3, or C10orf76 resulted in no discernible impact on the doublet pattern of endogenous CERT, with the hyperphosphorylated form being dominant (Fig. 3 A). This statement does not seem to correspond to the data in figure 3A. It looks to me that the unphosphorylated CERT band is more pronounced in the C10orf67 KO.

Author response: To address the concern about the phosphorylation state of CERT in the C10orf76 KO cells, we carried out a quantification analysis of the immunoblot images (Fig. 3A). As suggested, the hypo-phosphorylated form was increased by disruption of C10orf76. We further discussed the changes of phosphorylation state of CERT as follows at p. 16, l. 267: “Thus, there was the possibility that disrupting *PI4KB*, *ACBD3*, and/or *C10orf76* upregulated the SRM phosphorylation-dependent repression of CERT. However, *C10orf76* KO increased the ratio of the de- or hypo-phosphorylated form (i.e., active form) relative to the total CERT forms (Fig. 3 A). The change in the phosphorylation state of CERT in *PI4KB* KO was modest ($p = 0.07$), but the total amount of CERT was significantly increased (Fig. 3 A). These results eliminated the possibility that the gene disruption compromised the production of active CERT, thereby repressing the synthesis of SM. The results also imply that inappropriate dysfunction of CERT induced compensatory responses in the expression and phosphorylation state of the endogenous CERT in cells.”

Reviewer #3 (Comments to the Authors):

In this study, Mizuike et al. investigate the molecular mechanisms involved in ceramide transport from the ER to the Golgi by the ceramide transport protein CERT. They used a genome-wide CRISPR/Cas9 KO screen to identify genes for which KO confers Lysenin resistance. Lysenin binds to sphingomyelin at the plasma membrane and creates pores that lead to cell death. Trafficking of ceramide from the endoplasmic reticulum (ER) to the Golgi is required for sphingomyelin synthesis and, therefore, its trafficking to the plasma membrane. With this screen, the authors identified a number of genes for which KO confers lysenin resistance and that are thus potentially involved in ceramide trafficking. Among these genes, the authors focused on genes associated with PI4P regulation, which included PI4KB, a PI4P kinase that produces PI(4)P at the Golgi, and ACBD3 and C10orf76, which encode proteins interacting with PI4KB. The authors then tested whether these genes were required for the recruitment of the ceramide transfer

protein CERT at ER-Golgi contact sites. They first confirmed that PI4KB, ACBD3 and C10orf76 depletion confers lysenin resistance and that they are important for sphingomyelin synthesis. They then investigated whether these proteins were required for PI4KB and CERT localization and function to the Golgi. The authors then used super-resolution imaging to look for the localization of ACBD3 and C10orf76 at different Golgi compartments and observed that ACBD3 localizes more to proximal Golgi (Ci-Golgi) and that C10orf76 localizes more to distal Golgi (Trans-Golgi) and more precisely ER-Trans-Golgi contact sites where ceramide transfer and sphingomyelin synthesis occur.

This manuscript is clearly written, and the experiments were well described in text/figures. Also, the quality of the experiments was high quality. Moreover, this study shows an important role for the C10orf76-PI4KB axis that notably produces PI(4)P at the Golgi in sphingomyelin trafficking to the plasma membrane, which requires ER to Golgi ceramide transfer. This would be of interest to the readers of JCB.

However, I have some major concerns with some of the interpretations of their data made by the authors as alternative interpretations were not addressed, and for others, control experiments are missing. Together, these issues make it difficult to conclude that ACBD3 and C10orf76 are required for Golgi recruitment of PI4KB for CERT activation.

Major points:

1) The authors conclude that PI4KB, ACBD3 and C10orf76 are important for CERT localization at the Golgi and that ACBD3 and C10orf76 are important for PI4KB recruitment at the Golgi. However, the Golgi enrichment of CERT-mVenus (fig 3B-c) and Flag-PI4KB (fig 4B-c) does not show any impairment at basal conditions and only when their localization is being pharmacologically forced (Myriocin for CERT and PIK93 for PI4KB). One would expect to see a defect at basal condition if any of these proteins are required for CERT or PI4KB recruitment to the Golgi Apparatus. There are several possible reasons for this. i) First, looking closely at the GM130 staining, the Golgi Apparatus looks different between the various KO, with ACBD3-del appearing more condensed. In fact, the GM130 staining in Figure 4B appears to be missing in the PI4KB-del and C10orf76-del cells. Since GM130 was used as the template for measuring the Golgi apparatus area, the difference in the Golgi may affect the enrichment quantification. ii) In Figure 3A, the authors show that CERT expression was increased in the KO cell lines. Could this increase in endogenous CERT be completed with the ectopic

expression of CERT-mVenus? iii) CERT and PI4KB enrichment measurements are the ratio of the CERT-mVENUS or FLAG-PI4KB signal intensity in Golgi localized vs the cytosol. For this quantification, the assumption is that the ectopically expressed cells between the different cell lines have the same expression. Has this been quantified? To resolve some of these issues, the authors should use antibodies against endogenous proteins. I am not sure whether the CERT antibody use in 3A is feasible for IF, but the PI4KB should be feasible for these studies.

Author response: Thank you very much for your critical and constructive comments. The morphological changes of the Golgi complex in the KO cells were certainly a concern in image analysis, yet it is an inevitable problem. Therefore, to respond to the reviewer's comment i), we added an explanation (p. 17, l. 286) that "Note that it is possible that the morphological changes of the Golgi complex may have affected the enrichment quantification, making it difficult to detect the difference at basal condition." To respond to comment ii), we added the new data and an explanation as follows (p. 18, l. 300): "the expression levels and the phosphorylation state of endogenous CERT in the KO cells tended to be unaltered by ectopically expressing CERT-mVenus (Fig. S3 A, see the contrasted image)." To answer comment iii), we confirmed that the expression levels of CERT-mVenus and FLAG-PI4KB in the KO cells were similar to those in the parent cells (Fig. S3 A and B). We did attempt to use antibodies against endogenous CERT and PI4KB, but those we purchased were not feasible for IF.

2) For the CERT and PI4KB enrichment quantification, is GM130 a good marker given that PI4KB may be localized to the trans-Golgi? Further, another protein that localization at the Golgi should be used as a negative control to demonstrate and validate their quantification. Similarly, using Brefeldin A treatment which was shown to abolish PI4KB localization at the Golgi, should also be used as a control.

Author response: As pointed out, GM130 is a *cis*-Golgi marker, not a *trans*-Golgi marker. At conventional microscopic resolutions, it is practically impossible to distinguish the Golgi *cis* stack, *trans* stack, and TGN. Antibodies that can histochemically detect endogenous GM130 in human cells are commercially available, but there is no good antibody for an endogenous *trans*-Golgi stack marker. Therefore, we regarded GM130 as a representative marker of the Golgi complex in conventional microscopic analysis. We also revised the image analysis method of Golgi-enrichment analysis to use Pearson's

correlation coefficient. The previous method was in analogy with Manders colocalization coefficient, which did not evaluate the correlations of the proteins of interest and the marker protein, and that method may have overestimated the colocalization rate due to the cytoplasmic signals and/or the morphological changes of the Golgi complex. Of note, PI4KB was recruited to both *cis*- and *trans*-Golgi stacks in wild-type cells because PI4KB is capable of binding to both ACBD3 and C10orf76. This situation is illustrated in Figure 7. We also employed another marker, SMS1-V5, as possibly the best *trans*-Golgi stack marker despite it being an epitope-tagged ectopic protein. However, ectopic expression of SMS1-V5 resulted in an intense decrease of CERT-mVenus for unknown reasons, making it impossible to conduct a microscopic analysis.

3) Based on their western blot of CERT in Figure 3A, the author stated that the knockout had "...no discernible impact on the doublet pattern of endogenous CERT, with the hyperphosphorylation from being dominant." Although I would agree that there is a higher ratio of the hyperphosphorylation band compared to the lower band for most, this is not the case for C10orf76A-del, where the lower band appears higher. Wouldn't this be expected if C10orf76A is required for SM production, as seen in figure 2A? In fact, wouldn't you see less hyperphosphorylated band in all the KO cell lines, given that SM is lower?

Author response: Thank you for the insightful suggestion. We carried out a quantitative analysis of the immunoblot image showing the doublet bands of CERT (Fig. 3 A). As you have commented, the increase of active-form CERT (the lower band) by disruption of C10orf76 is likely to indicate the requirement of C10orf76 for SM production. Then, we discussed as follows (p. 17, l. 274): "The results also imply that inappropriate dysfunction of CERT induced compensatory responses in the expression and phosphorylation state of the endogenous CERT in cells."

4) One important control lacking for all the KO cell lines is a morphologic analysis of the Golgi to show that the overall Golgi morphology is not affected by the various knockouts or treatments in a way that would affect the results of colocalization measures.

Author response: We did not argue that Golgi morphology was not changed in the KO cells. Rather, we described that discernible fragmentation of the Golgi seemed to occur in the KO cells, especially in *ACBD3* KO cells, in agreement with previous studies (please

see also the second paragraph of our response to comment #5). We note the concern that such morphological changes may have affected colocalization measures. However, given that these morphological changes by disruption of *ACBD3* are inevitable in nature, we employed STED to partly overcome this limitation. Fortunately, STED enabled us to distinguish Golgi sub-compartments (namely, GM130-distributed stack, SMS1-distributed stack, and TGN46-distributed stack) in WT cells. Using STED analysis, we could show that CERT, SMS1, and C10orf76, but not ACBD3, were preferentially distributed to distal Golgi regions and had significant overlapping distribution patterns. Overall, we consider that the results of this study are sound and lead to the conclusion that the C10orf76–PI4KB axis causes CERT to function at the distal Golgi.

5) The authors provide very robust data indicating that deletion of PI4KB, C10orf76 or ACBD3 impairs sphingomyelin synthesis and levels at the plasma membrane but do not show controls that the lysenin resistance conferred by these deletions is primarily due to a defect in ceramide transfer and not sphingomyelin trafficking. Such controls should be performed. For instance, confirming the ER-to-Golgi ceramide trafficking defect by using fluorescent ceramide or ceramide antibodies would strengthen the conclusions. Similarly, validating sphingomyelin synthesis defects by using a sphingomyelin biosensor (equinatoxin) or antibodies would support the claims of the authors.

Author response: Thank you very much for your helpful comments. We performed an assay of intracellular trafficking of ceramide using C₅-DMB-ceramide (Fig. 2 E) and obtained new data indicating that ER-to-Golgi movement of the fluorescent ceramide was impaired in the *PI4KB* KO and *ACBD3/C10orf76* DKO cells. We added paragraphs at p. 14, l. 232: “To examine the suggestion further, we analyzed the intracellular movement of C₅-DMB-ceramide, a fluorescent analog of ceramide, which at least partially recapitulates the CERT-dependent ER-to-Golgi trafficking of natural ceramide in living cells (Fukasawa et al., 1999; Hanada et al., 2003). When cells were incubated with C₅-DMB-ceramide at 4°C for 30 min, intracellular reticular structures (i.e., the ER) were mainly labeled in all cell types examined (Fig. 2 E). After chasing the prelabelled cells at 37°C for 10 min, the fluorescent signals were redistributed to the perinuclear regions (i.e., the Golgi complex) in the parent cells, whereas the perinuclear redistribution of the fluorescence in the *PI4KB* SKO and *ACBD3/C10orf76* DKO cells was reduced to the level in the *CERT* KO cells (Fig. 2 E). As expected, the impact on

the Golgi-redistribution of C₅-DMB-ceramide was smaller in the *ACBD3* SKO and *C10orf76* SKO cells than in the DKO cells (Fig. 2 E). Collectively, these results showed that disrupting *PI4KB* and *C10orf76* (especially with co-disruption of *ACBD3*) impaired the CERT-dependent trafficking of ceramide from the ER to the Golgi site for *de novo* SM synthesis.

It should also be noted that the C₅-DMB-ceramide-enriched regions in the *ACBD3* SKO cells did not display a ribbon-like Golgi morphology, in line with a previous study showing that *ACBD3* knockdown caused the fragmentation of the Golgi apparatus (Liao et al., 2019). However, regions in *PI4KB* SKO, *C10orf76* SKO cells, and *ACBD3/C10orf76* DKO cells did appear to be ribbon-like (Fig. 3E). Although we do not know how *ACBD3* SKO cells, but not *PI4KB* SKO nor *ACBD3/C10orf76* DKO cells, exhibited a strong impact on Golgi morphology, it is possible that the unbalance in the Golgi-PtdIns(4)P distribution caused by disruption of *ACBD3* (which will be discussed later) may serve as a signal to trigger the Golgi fragmentation.”

To examine the activity of the *de novo* synthesis of SM, we think the metabolic labeling with radioactive serine is the best method, and the experimental data had been provided in the original version (Fig. 2 D). In addition, the lipidome data to verify the defect of SM synthesis are presented (Fig. 2 C, Fig. S2 C and D). Thus, we think it is not necessary to perform additional experiments to further verify SM synthesis defects by using an SM biosensor or antibodies.

Pertaining to the crucial comment that the lysenin resistance conferred by these deletions was primarily due to a defect in ceramide transfer and not sphingomyelin trafficking, we discussed this point as follows (p. 26, l. 435): “Metabolic labeling of lipids with radioactive serine and intracellular movement of C₅-DMB-ceramide provided compelling evidence that disrupting the C10orf76–PI4KB axis compromises both *de novo* synthesis of SM and ER-to-Golgi transport of ceramide in living cells (Fig. 2 D and E). Acquiring the lysenin-resistance by disrupting the C10orf76–PI4KB axis via compromising the function of CERT was in line with previous studies showing the dysfunction of CERT endows lysenin-resistance to cells (Hanada et al., 2003; Yamaji & Hanada, 2014; Murakami et al., 2021). However, we do not deny the possibility that disrupting the C10orf76–PI4KB axis might also have compromised the transport of SM from the distal Golgi to the PM in the present study.” In addition, we briefly explained why only partial loss of SM is enough to acquire considerable lysenin resistance (p. 13, l. 209): “Lysenin binds to clustered SM, not mono dispersed SM molecules, in membranes (reviewed in Yilmaz et al., 2018), and only a partial decrease in the cellular SM level is enough to gain discernible resistance to lysenin in mammalian cells (Hanada et al.,

1998).”

Minor points:

- Fig1B: If possible, a complementation assay performed with PI4KB mutants unable to interact with either ACBD3 or C10orf76 could be an interesting experiment to perform in order to support the different roles of these two proteins in PI4KB recruitment/function at the Golgi.

Author response: This is an interesting point, and we did attempt the complementation assay but were unable to adjust the expression levels of the PI4KB mutants equally. Therefore, we could not include the data in the manuscript.

- Fig3A: the authors should provide a quantification of the immunoblot comparing levels of CERT but also the ratio phosphorylated/non-phosphorylated of CERT in the different conditions. Especially since the state in the text that they did not observe any discernible effect on the doublet (with the upper band being dominant) pattern between conditions. The image shown here seems to indicate that the ratio between phosphor and non-phospho band changes between conditions and that maybe the lower band appears dominant in C10orf76 KO cells. The authors should quantify that and (if there is a change) comment/discuss it in the text.

Author response: We carried out a quantitative analysis of the immunoblot image showing the doublet bands of CERT (Fig. 3 A).

- Fig3C and other Golgi enrichment measures: It is not clear in the text, figure or legends what is the units of Golgi enrichment that are measured/shown here (arbitrary units (but how they were defined?), percentage of colocalization? etc.). This should be indicated in the figures and in the legends.

Author response: We have revised the image analysis method to use Pearson’s correlation coefficient and added an explanation in the figure and the legend as follows: “The dots represent the Pearson’s correlation coefficient between CERT-mVenus and the Golgi marker GM130, and the line segments represent the median.”

- Fig 3C: The authors should make it obvious that the statistical analysis was performed on Myriocin treated cells.

Author response: We have changed the color of the * and lines showing the statistical difference to the same color to represent the myriocin-treated or non-treated samples. We also added an explanation in the legend as follows: “Orange, myriocin-untreated; light blue, myriocin-treated. Statistical differences between the parent and KO cells in each of the myriocin-treated and untreated conditions are shown. *, $p < 0.05$; ***, $p < 0.0005$.”

- Fig 3D: insets are particularly small; it would help the reader if their size is increased.

Author response: We have enlarged the size of the insets.

- Fig4A: the blot should be quantified as well.

Author response: We quantified the band intensity (Fig. 4 A) and added a description as follows (p. 19, l. 314): “Notably, we found that gene disruption of *ACBD3* and *C10orf76* did not affect the protein expression level of PI4KB (Fig. 4 A), although double disruption led to a modest decrease ($p = 0.07$).”

- Fig4D: The fact that only the P values below 0.05 are shown that all other pairs are $p < 0.05$ does not appear clear on the graph. This should be modified to make it more obvious what is considered statistically significant or not.

Author response: We have modified the figure to show the statistical difference between the samples.

- Fig5C is particularly dim compared to other panels. Brightness should be increased to facilitate visualization by readers.

Author response: Thank you for the suggestion. We have increased the brightness for better visualization.

- Fig6a: the legend refers to Figure S5 for the uncropped image, but only 3 supplemental figures were provided.

Author response: Thank you for pointing this out. We have provided the uncropped

image as “Source Data”.

- Line 312-314: the authors refer to Fig4C, while it is Fig4D that should be referred to.

Author response: Thank you for pointing this out. We have corrected it.

- Line 461: It appears that one word is missing in this sentence. In its current version, this sentence does not mean anything.

Author response: We have revised the sentence to “The diversity of PtdIns(4)P molecular species (i.e., acyl chain profile) and the degree of polar-headgroup density in the membrane may confer distinct properties on the local membrane environment.” (p. 31, l. 517)

Data only for the Reviewers

P4M-SidM-mVenus expressed in HeLa parent cells

inverted image

P4M-SidM-mVenus expressed in PI4KB KO cells

inverted image

The perinuclear punctate structures, to which P4M-SidM-mVenus was accumulated (indicated by arrowheads), probably represent the Golgi complex.

Inverted image: inverted the lookup table of each image for better visualization.

Many vacuolar structures (indicated by arrows) appeared in cells when P4M-SidM-mVenus was expressed. This is unlikely a non-specific event by expressing a mVenus-fusion protein because such vacuolar structures did not appear in the cells expressing CERT-mVenus (see Fig. 3 B of the manuscript).

February 3, 2023

Re: JCB manuscript #202111069RR

Dr. Kentaro Hanada
National Institute of Infectious Diseases
Department of Quality Assurance, Radiation Safety & Information System
1-23-1, Toyama, Shinjuku-ku
Tokyo 162-8640
Japan

Dear Dr. Hanada,

Thank you for submitting your revised manuscript entitled "The C10orf76-PI4KB axis orchestrates CERT-mediated ceramide trafficking to the distal Golgi". The manuscript has been seen by the original reviewers whose full comments are appended below. While the reviewers continue to be largely positive about the work in terms of its suitability for JCB, some important issues remain.

You will see that reviewers #1 and #3 feel that that some of the conclusions, especially the idea that the C10orf76-PI4KB axis is a major player in the modulation of CERT localization, remain insufficiently supported. Both reviewers have offered suggestions for improving the manuscript, including toning down some of the statements (see reviewer #1's points) and the addition of more complete quantifications of the data and thorough statistical analyses (reviewer #3). Therefore, we will need to see these concerns addressed in full before we can proceed with publication.

Our general policy is that papers are considered through only one revision cycle; however, given that the suggested changes are fairly straightforward, we are open to one additional short round of revision. Please note that we will expect to make a final decision without additional reviewer input upon resubmission.

Please submit the final revision within one month, along with a cover letter that includes a point by point response to the remaining reviewer comments.

Thank you for this interesting contribution to Journal of Cell Biology. You can contact me or the scientific editor listed below at the journal office with any questions, cellbio@rockefeller.edu.

Sincerely,

Tamas Balla, MD, PhD
Monitoring Editor
Journal of Cell Biology

Tim Spencer, PhD
Executive Editor
Journal of Cell Biology

Reviewer #1 (Comments to the Authors (Required)):

My concerns have largely been addressed. However, I think the major claims should be substantially toned down. For example, the discussion states: "the C10orf76-PI4KB axis is crucial for producing PtdIns(4)P at the distal Golgi regions." This strong claim is not supported by the data. At best, one can say the axis may contribute to producing PtdIns(4)P. In the rebuttal letter, the authors say there are "technical problems in observing PtdIns(4)P by super-resolution microscopy." Considering that, the case that PtdIns(4)P is specifically enriched on the CERT-positive Golgi regions is not strong. Moreover, PI4KB, ACBD3, and C10orf76 make only a modest contribution to CERT enrichment at the Golgi. Figure 3C show that knock down of these genes causes the Pearson's correlation of CERT and a Golgi marker to decrease from ~0.65 to 0.5-0.55 (without myriocin). This suggest that the C10orf76-PI4KB axis contributes to CERT localization but seems unlikely to be the dominant factor. This should be acknowledged and discussed.

Reviewer #2 (Comments to the Authors (Required)):

In this revised version the authors have responded as best as they could to my criticisms. I think that the identification of the C10orf67 axis in the transfer of ceramide by CERT to synthesize SM is important. Unfortunately, they do not have a detailed mechanism of where the specificity is determined, but they have experiments that provide evidence against some possible mechanisms, like coincidence detection of PI4P and C10orf67. The manuscript represents a lot of work already so my suggestion would be not ask for this additional mechanistic insight. Hopefully, it will come in a future manuscript.

Reviewer #3 (Comments to the Authors (Required)):

The authors addressed most of my concerns. However, the role of C10orf76 and ACBD3 in CERT-mediated ceramide trafficking and their subcellular localization is still not convincing. The main issue is the lack of robust quantification that is required to interpret the experiments for the two experiments stated above.

To address whether ACBD3 or C10orf76 is required for ceramide transport from the ER to Golgi, the authors visualized the trafficking of cold-loaded C5-DMB-Cer in a chase experiment (Fig 2E). To show Golgi targeting or lack thereof, the authors measured the fluorescent intensity of the perinuclear region using a line scan. The authors claim a decrease in fluorescent intensity for the PI4KB SKO and ACBD3/C10orf76 DKO compared to the control. However, as only two cells are shown per cell line with only a single line scan quantification, one cannot make such conclusions from these data. More robust quantification of DMB trafficking is needed. Therefore, the authors should perform a proper quantification (e.g. Pearson's) on an appropriate number of cells using a Golgi marker (GM130-GFP or Cellight Golgi GFP or equivalent). Without such quantification, it is difficult to compare any differences between the knockout cells.

Similarly, Figures 5, 6 and S4 require a more robust quantification of Manders M1 and M2. At present, there is only quantification of a single image. A quantification of a single Golgi image is not appropriate. A more robust quantification is required. Manders quantification of GM130-TGN46 should be used as a control.

Finally, missing in the statistical analysis is the number of trials, and population size is not given, making it difficult to assess the data. Please add such information in the legends.

minor points

- Molecular weight indicators are required for 3A, 4A, S2B, and 4E

Dear Dr. Balla, Editor of the Journal of Cell Biology,

We appreciate the time and effort that you and the reviewers have dedicated to providing feedback on our manuscript and are grateful for the insightful comments. We have incorporated most of the suggestions from the editor and reviewers. These changes have been highlighted within the manuscript. In addition, we also added the following sentence at the end of the discussion: "C10orf76 is a typical name to refer to hypothetical or predicted genes that have not yet been characterized although it has been predicted to contain an Armadillo repeat structure. Since this study demonstrated that C10orf76 acts as a mediator to recruit PI4KB to the distal Golgi regions, we would like to propose DGARM as an alternative name for C10orf76 after a distal Golgi Armadillo repeat protein." Below, please find our responses to the reviewers' comments, addressed in a point-by-point manner.

I hope that our revised manuscript appropriately addresses the issues raised by the reviewers.

Sincerely,

Reviewer #1 (Comments to the Authors (Required)):

Comment: My concerns have largely been addressed. However, I think the major claims should be substantially toned down. For example, the discussion states: "the C10orf76-PI4KB axis is crucial for producing PtdIns(4)P at the distal Golgi regions." This strong claim is not supported by the data. At best, one can say the axis may contribute to producing PtdIns(4)P. In the rebuttal letter, the authors say there are "technical problems in observing PtdIns(4)P by super-resolution microscopy." Considering that, the case that PtdIns(4)P is specifically enriched on the CERT-positive Golgi regions is not strong.

Thank you very much for the scientifically careful comment. As the reviewers pointed out, we showed distal Golgi distribution of C10orf76, CERT, and SMS1, but not PtdIns(4)P, using STED analysis. Since lipids embedded in membranes are not completely immobilized even after the glutaraldehyde fixation (Tanaka et al., Nature Methods), it is practically infeasible to precisely determine the intra-Golgi distribution of PtdIns(4)P at the STED resolution level as described in the previous rebuttal letter. Thus, we revised the previous sentences as follows: "Here, we found that ACBD3 and C10orf76 act as mediators to recruit PI4KB to different Golgi regions. Additionally, the C10orf76-PI4KB axis was found to play a role in producing

PtdIns(4)P for the recruitment of CERT to the distal Golgi regions. Once there, CERT facilitates the transfer of ceramide from the ER for the production of SM.” (L435-433). We also briefly explained why STED analysis of PtdIns(4)P may encounter an inevitable technical concern as follows: “Lipid molecules were shown to be discernibly mobile even after fixation by glutaraldehyde or formaldehyde (Tanaka et al., 2010), not allowing us to analyze the intracellular distribution of PtdIns(4)P at the STED super-resolution level. ” (L402-405). In addition, the sentence “CERT predominantly utilizes PtdIns(4)P”, in the Abstract, was also toned down to “CERT preferentially utilizes PtdIns(4)P” (L23).

Comment: Moreover, PI4KB, ACBD3, and C10orf76 make only a modest contribution to CERT enrichment at the Golgi. Figure 3C show that knock down of these genes causes the Pearson's correlation of CERT and a Golgi marker to decrease from ~0.65 to 0.5-0.55 (without myriocin). This suggest that the C10orf76-PI4KB axis contributes to CERT localization but seems unlikely to be the dominant factor. This should be acknowledged and discussed.

To answer this comment, we added more explanation in the discussion (L439-459) as follows: “Metabolic labeling of lipids with radioactive serine and intracellular movement of C₅-DMB-ceramide provided compelling evidence that disrupting the C10orf76–PI4KB axis compromises both *de novo* synthesis of SM and ER-to-Golgi transport of ceramide in living cells (Fig. 2 D and E), that are likely due to the dysfunction of CERT. Clear impacts of the C10orf76 disruption on the Golgi-localization of CERT were observed upon myriocin treatment (Figure 3 B and C), although the Pearson's correlation values of CERT-mVenus and the Golgi marker only marginally differed between parental cells and C10orf76 KO cells in the absence of myriocin. This may be at least partially accounted for by a compensatory increase of the active form of CERT-mVenus in the C10orf76 KO cells (Fig. S3 A): since the difference in the conformation of CERT is likely to affect the subcellular distribution. Myriocin treatment increased the dephosphorylated form of CERT to almost the same extent in all five cell lines examined (Figure S3 A), allowing us to examine the effects of the gene disruptions on CERT localization under the conditions where gene disruption-dependent alterations of the phosphorylation states of CERT-mVenus were minimized (Figure 3 C). Collectively, these results indicated that the C10orf76–PI4KB axis contributes to the recruitment of CERT to the Golgi complex. Acquiring the lysenin-resistance by disrupting the C10orf76–PI4KB axis via compromising the function of CERT was in line with previous studies showing that the dysfunction of CERT endows lysenin-resistance to cells (Hanada et al., 2003; Yamaji and Hanada, 2014; Murakami et al., 2021). However, we do not deny the possibility that disrupting the C10orf76–PI4KB axis might also have compromised the transport of SM from the distal

Golgi to the PM in the present study.” We also added “C10orf76 is a typical name to refer to hypothetical or predicted genes that have not yet been characterized although it has been predicted to contain an Armadillo repeat structure. Since this study demonstrated that C10orf76 acts as a mediator to recruit PI4KB to the distal Golgi regions, we would like to propose DGARM as an alternative name for C10orf76 after a distal Golgi Armadillo repeat protein.” in the last of the discussion.

Reviewer #2 (Comments to the Authors (Required)):

Comment: In this revised version the authors have responded as best as they could to my criticisms. I think that the identification of the C10orf67 axis in the transfer of ceramide by CERT to synthesize SM is important. Unfortunately, they do not have a detailed mechanism of where the specificity is determined, but they have experiments that provide evidence against some possible mechanisms, like coincidence detection of PI4P and C10orf67. The manuscript represents a lot of work already so my suggestion would be not ask for this additional mechanistic insight. Hopefully, it will come in a future manuscript.

Thank you very much for your evaluation.

Reviewer #3 (Comments to the Authors (Required)):

Comment: The authors addressed most of my concerns. However, the role of C10orf76 and ACBD3 in CERT-mediated ceramide trafficking and their subcellular localization is still not convincing. The main issue is the lack of robust quantification that is required to interpret the experiments for the two experiments stated above.

To address whether ACBD3 or C10orf76 is required for ceramide transport from the ER to Golgi, the authors visualized the trafficking of cold-loaded C5-DMB-Cer in a chase experiment (Fig 2E). To show Golgi targeting or lack thereof, the authors measured the fluorescent intensity of the perinuclear region using a line scan. The authors claim a decrease in fluorescent intensity for the PI4KB SKO and ACBD3/C10orf76 DKO compared to the control. However, as only two cells are shown per cell line with only a single line scan quantification, one cannot make such conclusions from these data. More robust quantification of DMB trafficking is needed. Therefore, the authors should perform a proper quantification (e.g. Pearson's) on an appropriate number of cells using a Golgi marker (GM130-GFP or Cellight Golgi GFP or equivalent). Without such quantification, it is difficult to compare any differences between the knockout cells.

Thank you very much for the constructive comments. Neither GM130-GFP nor Cellight Golgi GFP are adaptable for colocalization analysis with DMB-ceramide for their overlapped fluorescence properties. In order to respond to the comment, we therefore employed a commercial Golgi marker plasmid pDsRed-monomer-Golgi (clontech, 632480) for the DMB-ceramide trafficking assay. However, we found a problem for using the plasmid as a marker for image analysis. When transiently expressed, a considerable number of cells showed dispersed DsRed signals, which resembles those of the ER (see the figure for editor/reviewer-only), not allowing us to use the DsRed signal as a Golgi marker under our experimental conditions. Since fluorescent ceramide analogues are discernibly mobile even after fixation by glutaraldehyde or formaldehyde (Pagano et al., *J. Cell Biol.*, (1989), **190**:2067-2079), prompt observation of these analogues after fixation is essential. Thus, it is technically infeasible to employ immunostaining with antibodies against an appropriate endogenous Golgi marker (i.e., GM130 or TGN46) for colocalization assay with DMB-ceramide. Due to these technical limitations, we could not obtain new results with publishable quality, and therefore rewrote the term “Golgi-redistribution” to “perinuclear redistribution” for the explanation of the result of the DMB-ceramide assay. In addition, in the result of the DMB-ceramide assay (without DsRed-Golgi), we increased the line scan quantification of the perinucleus regions ($n > 25$ cells) to show the mean \pm SD of line profiles (Fig. 2 E).

Comment: Similarly, Figures 5, 6 and S4 require a more robust quantification of Manders M1 and M2. At present, there is only quantification of a single image. A quantification of a single Golgi image is not appropriate. A more robust qualification is required. Manders quantification of GM130-TGN46 should be used as a control.

Thank you for the pivotal suggestion. To answer this comment, we carried out additional quantification ($n = 6-12$ cells) and showed the mean \pm SD of Manders correlation coefficient (MCC) M1 and M2 (Fig. 5 B and D, Fig. S4 C and D). We also quantified the MCCs of GM130-TGN46 as a control (Fig. S4 B and D).

Comment: Finally, missing in the statistical analysis is the number of trials, and population size is not given, making it difficult to assess the data. Please add such information in the legends.

Thank you for pointing out. We added the information in the legends.

minor points

- Molecular weight indicators are required for 3A, 4A, S2B, and 4E

We added the weight indicators in the figures.

Data only for the Editors/Reviewers

We attempted to use the pDsRed-monomer-Golgi plasmid (clontech, 632480) for transiently expressing the fluorescence-conjugated Golgi marker. However, a substantial portion of cells showed a dispersed DsRed signals (white arrows) both in the 0 min- and 10 min-chased cells. (Representative images of the parent HeLa cells are shown.) Therefore we concluded that this DsRed-Golgi is not suitable for quantitative image analysis. Note that the DMB signal was not leaked into the Red channel even at the condensed perinuclear regions in the 10 min-chased cells.

Overnight cultured cells were transfected with pDsRed-monomer-Golgi plasmid (0.5 μ g / well, 24-well plate) and incubated for 24 h. Cells were labeled with C₅-DMB-ceramide complexed with BSA at 4°C for 30 min and chased at 37°C for 0 or 10 min. Cells were fixed and subsequently observed by fluorescent microscopy.

March 17, 2023

RE: JCB Manuscript #202111069RRR

Dr. Kentaro Hanada
National Institute of Infectious Diseases
Department of Quality Assurance, Radiation Safety & Information System
1-23-1, Toyama, Shinjuku-ku
Tokyo 162-8640
Japan

Dear Dr. Hanada:

Thank you for submitting your revised manuscript entitled "The C10orf76-PI4KB axis orchestrates CERT-mediated ceramide trafficking to the distal Golgi". We have now assessed your revised paper and we would be happy to publish your paper in JCB pending final revisions necessary to meet our formatting guidelines (see details below).

A. MANUSCRIPT ORGANIZATION AND FORMATTING:

- 1) Text limits: Character count for Articles is < 40,000, not including spaces. Count includes the abstract, introduction, results, discussion, and acknowledgments. Count does not include title page, materials and methods, figure legends, references, tables, or supplemental legends. Your paper is currently below this limit but we ask you to bear it in mind when revising.
- 2) Figure formatting: Scale bars must be present on all microscopy images, including inset magnifications. Molecular weight or nucleic acid size markers must be included on all gel electrophoresis.
- 3) Statistical analysis: Error bars on graphic representations of numerical data must be clearly described in the figure legend. The number of independent data points (n) represented in a graph must be indicated in the legend. Statistical methods should be explained in full in the materials and methods. For figures presenting pooled data the statistical measure should be defined in the figure legends. Please also be sure to indicate the statistical tests used in each of your experiments (both in the figure legend itself and in a separate methods section) as well as the parameters of the test (for example, if you ran a t-test, please indicate if it was one- or two-sided, etc.). Also, if you used parametric tests, please indicate if the data distribution was tested for normality (and if so, how). If not, you must state something to the effect that "Data distribution was assumed to be normal but this was not formally tested." It seems like all the tests you used were non-parametric but please confirm this.
- 4) Materials and methods: Should be comprehensive and not simply reference a previous publication for details on how an experiment was performed. Please provide full descriptions (at least in brief) in the text for readers who may not have access to referenced manuscripts. The text should not refer to methods "...as previously described."
- 5) Please be sure to provide the sequences for all of your primers/oligos and RNAi constructs in the materials and methods. You must also indicate in the methods the source, species, and catalog numbers (where appropriate) for all of your antibodies.
- 6) Microscope image acquisition: The following information must be provided about the acquisition and processing of images:
 - a. Make and model of microscope
 - b. Type, magnification, and numerical aperture of the objective lenses
 - c. Temperature
 - d. imaging medium
 - e. Fluorochromes
 - f. Camera make and model
 - g. Acquisition software
 - h. Any software used for image processing subsequent to data acquisition. Please include details and types of operations involved (e.g., type of deconvolution, 3D reconstitutions, surface or volume rendering, gamma adjustments, etc.).
- 7) References: There is no limit to the number of references cited in a manuscript. References should be cited parenthetically in the text by author and year of publication. Abbreviate the names of journals according to PubMed.

- 8) Supplemental materials: There are strict limits on the allowable amount of supplemental data. Articles may have up to 5 supplemental figures. At the moment, you are below this limit but please bear it in mind when revising. Please also note that tables, like figures, should be provided as individual, editable files. A summary of all supplemental material (that is, in addition to the supplementary figure legends) should appear at the end of the Materials and methods section. Please see any recent JCB paper for an example of this. ****Also, we recommend renaming your "supplemental document" to "supplemental code". Please be sure to also rename the callout on page 45 and the title on page 58.****
- 9) eTOC summary: A ~40-50 word summary that describes the context and significance of the findings for a general readership should be included on the title page. The statement should be written in the present tense and refer to the work in the third person. It should begin with "First author name(s) et al..." to match our preferred style.
- 10) Conflict of interest statement: JCB requires inclusion of a statement in the acknowledgements regarding competing financial interests. If no competing financial interests exist, please include the following statement: "The authors declare no competing financial interests." If competing interests are declared, please follow your statement of these competing interests with the following statement: "The authors declare no further competing financial interests."
- 11) A separate author contribution section is required following the Acknowledgments in all research manuscripts. All authors should be mentioned and designated by their first and middle initials and full surnames. We encourage use of the CRediT nomenclature (<https://casrai.org/credit/>).
- 12) ORCID IDs: ORCID IDs are unique identifiers allowing researchers to create a record of their various scholarly contributions in a single place. At resubmission of your final files, please consider providing an ORCID ID for as many contributing authors as possible.
- 13) Journal of Cell Biology now requires a data availability statement for all research article submissions. These statements will be published in the article directly above the Acknowledgments. The statement should address all data underlying the research presented in the manuscript. Please visit the JCB instructions for authors for guidelines and examples of statements at (<https://rupress.org/jcb/pages/editorial-policies#data-availability-statement>).
- 14) Please check your source data for supplemental figure 3 - it appears to us that the bands with the box around it in the source data file are not the same bands presented in the figure. Instead, it looks like you used replicate #3 (the far right set of bands) from the gel. Of course, that is perfectly fine but please place the box around the correct bands in the source data file when submitting your revised files.

B. FINAL FILES:

****It is JCB policy that if requested, original data images must be made available to the editors. Failure to provide original images upon request will result in unavoidable delays in publication. Please ensure that you have access to all original data images prior to final submission.****

****The license to publish form must be signed before your manuscript can be sent to production. A link to the electronic license to publish form will be sent to the corresponding author only. Please take a moment to check your funder requirements before choosing the appropriate license.****

Thank you for your attention to these final processing requirements. Please revise and format the manuscript and upload

materials within 7 days. If complications arising from measures taken to prevent the spread of COVID-19 will prevent you from meeting this deadline (e.g. if you cannot retrieve necessary files from your laboratory, etc.), please let us know and we can work with you to determine a suitable revision period.

Please contact the journal office with any questions, cellbio@rockefeller.edu.

Thank you for this interesting contribution, we look forward to publishing your paper in Journal of Cell Biology.

Sincerely,

Tamas Balla, MD, PhD
Monitoring Editor
Journal of Cell Biology

Tim Spencer, PhD
Executive Editor
Journal of Cell Biology